# Calibrating 1D hydrodynamic river models in the absence of cross-section geometry using satellite observations of water surface elevation and river width

Liguang Jiang[1, 2], Silja Westphal Christensen[2, 3], Peter Bauer-Gottwein[2]

[1]School of Environmental Science and Engineering, Southern University of Science and Technology, Shenzhen, 518055, China
[2]Department of Environmental Engineering, Technical University of Denmark, 2800 Kgs. Lyngby, Denmark
[3]Department of Applied Mathematics and Computer Science, Technical University of Denmark, 2800 Kgs. Lyngby, Denmark

*Correspondence to*: Liguang Jiang (jianglg@sustech.edu.cn)

**Abstract.** Hydrodynamic modeling has been increasingly used to simulate water surface elevation which is important for flood prediction and risk assessment. Scarcity and inaccessibility of in-situ bathymetric information have hindered hydrodynamic model development at continental-global scales. Therefore, river cross-section geometry is commonly approximated by highly simplified generic shapes. Hydrodynamic river models require both bed geometry and roughness as input parameters. Simultaneous calibration of shape parameters and roughness is difficult, because often there are trade-offs between them. Instead of parameterizing cross-section geometry and hydraulic roughness separately, this study introduces a parameterization of 1D hydrodynamic models by combining cross-section geometry and roughness into one conveyance parameter. Flow area and conveyance are expressed as power-laws of flow depth, and they are found to be linearly related in log-log space at reach scale. Data from a wide range of river systems show that the linearity approximation is globally applicable. Because the two are expressed as power-laws of flow depth, no further assumptions about channel geometry are needed. Therefore, the hydraulic inversion approach allows for calibrating flow area and conveyance curves in the absence of direct observations of bathymetry and hydraulic roughness. The feasibility and performance of the hydraulic inversion workflow are illustrated using satellite observations of river width and water surface elevation in the Songhua River, China. Results show that this approach is able to reproduce water level dynamics with root mean square error value of 0.44 m and 0.50 m at two gauging stations, which is comparable to that achieved using a standard calibration approach. In summary, this study puts forward an alternative method to parameterize and calibrate river models using satellite observations of river width and water surface elevation.

# 1 Introduction

Hydrodynamic modeling of rivers is important for quantitative assessment of river flow and water level dynamics and, critically, for risk assessment and flood prediction. It is widely used for many applications, such as estimates of hydraulic parameters (e.g. water surface elevation (WSE), longitudinal profile, velocity), flood forecasting, inundation estimation, risk assessment, river maintenance, etc. (Andreadis and Schumann, 2014; Bates et al., 2014; Bierkens, 2015; Blöschl et al., 2015; Jiang et al., 2020). Nowadays, in the era of big data, earth observation datasets, cloud computing, and complex modeling

platforms are available for better simulations of WSE at multiple scales (Fleischmann et al., 2019; Gleason and Durand, 2020; Ward et al., 2015).

Traditional hydrodynamic modeling approaches require a detailed river channel bathymetry, which is usually represented by a set of cross-section shapes, distributed along the river reach of interest. There are, however, only a limited number of rivers, for which the surveyed geometry is available. The challenge that arises in many studies is how to

approximate the channel geometry. This is a common problem which the scientific community faces. A common approach is to parameterize channel geometry as a simple shape, e.g. a rectangle or triangle (Garambois et al., 2017; Jiang et al., 2019; Neal et al., 2012; Schneider et al., 2017). Instead of rectangular or triangular shapes, Dingman (2007) and Neal et al. (2015) used a power function (bankfull width and depth are required) to represent channel shape variability between the limiting cases of rectangular and triangular shape. However, Neal et al. (2015) used a cross-section geometry which did not vary

along the channel. Similar parameterizations of cross-section shapes were used in Mejia & Reed (2011), and the effect of assumed shapes on simulated flows was investigated. Some studies estimated river bathymetry using global DEMs combined with an assumed simplified shape (e.g. rectangle) of the submerged portion of the river. Domeneghetti (2016) used DEM data to infer the river bathymetry based on width-elevation relationships of high flow and low flow, respectively. Similarly, a few studies infer bathymetry from water surface height and width by fitting the relationship between the two. Obviously, the

success of this approach depends on the channel exposure (Mersel et al., 2013). Moreover, combinations of remote sensing data and empirical statistical relationships or data assimilation approaches have also been used to infer effective bathymetry (Brisset et al., 2018; Dey et al., 2019; Durand et al., 2008; Fonstad and Marcus, 2005; Grimaldi et al., 2018; Larnier et al., 2020; Legleiter, 2015; Moramarco et al., 2019; Schaperow et al., 2019). For instance, Durand et al. (2008) estimated bathymetric depth and slope by assimilating synthetic WSE data from the Surface Water and Ocean Topography (SWOT)

mission into the LISFLOOD-FP hydrodynamic model. Larnier et al. (2020) also applied data assimilation to infer effective bathymetry from synthetic SWOT altimetry measurements within an inverse framework. Here, we do not comprehensively review bathymetry estimation using upcoming SWOT mission data. Instead, we refer the reader to Biancamaria et al. (2016) and Gleason & Durand (2020) for a broader overview.

In addition to the channel bathymetry, channel roughness is another factor that is important for simulating flow

dynamics with sufficient accuracy (Bates et al., 2014; Neal et al., 2015). Usually, a uniform value is adopted to represent channel/floodplain roughness although large heterogeneity of river roughness exists in most cases (Annis et al., 2020; Jiang

et al., 2020; Pappenberger et al., 2007; Schumann et al., 2007). When calibrating channel geometry parameters along with roughness parameters, strong parameter correlation appears between cross-section shape (wetted perimeter) and hydraulic roughness (Jiang et al., 2019). That is, the roughness parameter will be "effective", not only representing the friction but also compensating for inaccurate geometry, which affects the hydraulic resistance through the wetted perimeter. Therefore, roughness and geometry parameters trade-off against each other, which has been widely reported (see Garambois & Monnier (2015) and references therein).

In order to reduce parameter correlation in hydraulic inverse problems, we put forward a method to parameterize and calibrate 1-D river models in a different way. Instead of roughness and geometry, flow area and conveyance curves as functions of flow depth are estimated in an inverse modeling workflow. In this way, only the dependence of area and conveyance on flow depth is estimated, regardless of the detailed channel shape and roughness. This paper illustrates this approach for the calibration of a 1D MIKE Hydro River model (DHI, 2017) to simulate WSE dynamics, using satellite observations of WSE and river width. The novelty is to use power-law relationships between flow area / conveyance and flow depth in a hydraulic inversion without detailed cross-section data or assumption of any specific cross-section shape. Therefore, this approach is fundamentally different from previous studies, and provides an alternative way for hydrodynamic model calibration.

## 2 Methods

### 2.1 Theoretical background

Flow in open channels can be described by the continuity equation and momentum equation, known as the de Saint-Venant equations (Chow, 1959):

$$\frac{\partial A(d)}{\partial t} + \frac{\partial Q}{\partial x} = 0, \tag{1}$$

$$\frac{\partial Q}{\partial t} + \frac{\partial}{\partial x}\left(\frac{Q^2}{A(d)}\right) + gA(d)\frac{\partial d}{\partial x} - gA(d)\left(S_0 - S_f(d)\right) = 0, \tag{2}$$

where: $A$ is the cross-section area; $Q$ is the discharge; $d$ is the flow depth; $S_0$ is the slope of the channel bottom; $S_f$ is the friction slope; $g$ is the gravity acceleration; $t$ is time and $x$ is chainage, i.e. the distance along the channel.

Equations (1) and (2) compose the 1-D dynamic wave model. In the absence of cross-section geometry, there are five unknowns in this model, i.e. two variables ($Q$ and $d$) and three unknown values (A, $S_o$, and $S_f$), which are functions of further parameters as specified below. To solve for $Q$ and $d$, information about channel geometry and friction slope is required. Flow area $A$ and channel slope $S_o$ can be obtained once the bathymetry is known. The friction slope $S_f$ can be approximated using the Manning formula or the Chézy formula (Chow, 1959).

Here, we express friction slope as a function of conveyance (K) and discharge (Q) using Manning's equation

$$Q = K S_f^{\frac{1}{2}}, \tag{3}$$

$$K = \frac{1}{n} A R^{\frac{2}{3}}, \tag{4}$$

where, $n$ is the Manning roughness coefficient and $R$ is the hydraulic radius. The conveyance is a measure of water carrying capacity of a cross-section (Chow, 1959).

95       Substituting for $S_f$, the momentum equation is written as:

$$\underbrace{\underbrace{\frac{\partial Q}{\partial t} + \frac{\partial}{\partial x}\left(\frac{Q^2}{A(d)}\right)}_{inertia\ terms} + \underbrace{gA(d)\frac{\partial d}{\partial x} - \underbrace{gA(d)\left(S_0 - \frac{Q^2}{K^2(d)}\right)}_{Kinematic\ wave}}_{Diffusive\ wave}}_{Dynamic\ wave} = 0. \tag{5}$$

This version of the momentum Eq. (5) indicates that, in steady state (for both kinematic wave and diffusive wave), the calibration is much more sensitive to $K(d)$ than to $A(d)$, and $A(d)$ appears only when the flow accelerates or decelerates.

## 2.2 Parameterization of flow area and conveyance curves

Equations (1) and (5) are two equations with still five unknowns, i.e. two variables ($Q$ and $d$), and three unknown values ($A$, $S_o$, and $K$). However, $K$ and $A$ are related to flow depth, $d$. If $K$ and $A$ can be expressed as functions of $d$, $Q$ and $d$ can be solved for, given the slope $S_o$ but without the need for detailed information on cross-section shape and roughness. The hydraulic geometry relations are widely used to relate the water surface width, average depth, and average velocity to discharge since it was introduced by Leopold and Maddock in 1953 (Bjerklie et al., 2005; Dingman, 2007; Ferguson, 1986;

Gleason, 2015; Leopold and Maddock, 1953). Dingman (2007) has derived explicit equations for the exponent and coefficients in the power-law function, explaining the variation of hydraulic geometry in different rivers. In some way analogous to the at-a-station power-law of hydraulic geometry, power-laws that relate flow area $A$ and conveyance $K$ to flow depth $d$ of a cross-section can be written, respectively, as (Chow, 1959; Garbrecht, 1990):

$$A(d) = a\, d^{\beta}, \tag{6}$$

$$K(d) = c\, d^{\delta}, \tag{7}$$

$$d = H - Z_0, \tag{8}$$

where, $a$, $\beta$, $c$ and $\delta$ are empirical coefficients; $H$ and $Z_0$ are WSE and channel datum, i.e. water surface elevation for zero flow.

      Transforming Eq. (6) and Eq. (7) into log-log space, we can write the following linear relationships:

$$\log A(x,t) = \alpha(x) + \beta(x)\log d(x,t), \tag{9}$$

$$\log K(x,t) = \gamma(x) + \delta(x)\log d(x,t), \tag{10}$$

where $\alpha = \log(a)$ and $\gamma = \log(c)$. This relationship is investigated for several rivers to show its validity for real-world rivers. The width ranges between the rivers over three orders of magnitude. Note that, these six rivers are used simply due to the availability of cross-section data (see a map of rivers and cross-sections in Figure A1). Strong positive linear relationships are revealed by plotting the logarithmic $A \sim d$, and $K \sim d$ pairs for any given cross-section below bankfull depth (Figure 1). A discontinuity may occur if significant flood plain exists as the case of the Yellow River (Figure 1d). Chow (1959) and Garbrecht (1990) suggested using separate functions to approximate the hydraulic properties below and above bankfull depth. In this initial study, one single power-law is used. Note that the conveyance changes with the Manning's coefficient, but the linear relationship holds (Figure A2). To calculate conveyance, spatially varying, randomly distributed Manning's coefficient ranging between 0.015 and 0.05 are used to mimic real-world rivers instead of unrealistic uniform values along the whole reach. A uniform Manning's coefficient results in a much stronger linear relationship (Figure A2 and Figure A3).

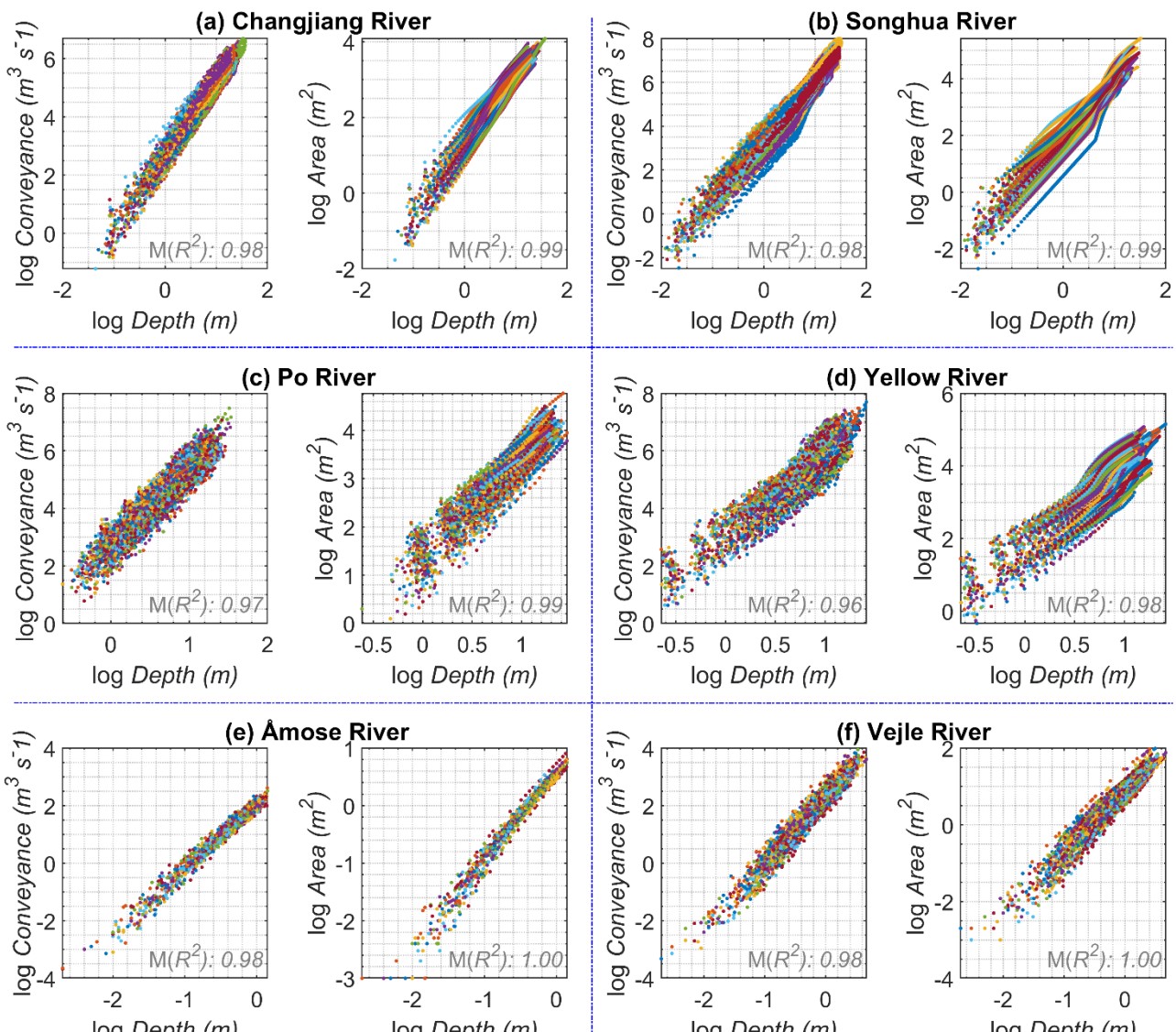

**Figure 1.** Plots of flow area and conveyance against flow depth in log-log space. In each plot, dots in the same color are from one certain cross-section. Linear relationships between logarithmic area / conveyance and depth are estimated for each cross-section, i.e. in an "at-a-station" manner. The median value of slopes of linear regression is given in each plot. In total, there are 60, 70, 335, 98, 51, 165 cross-sections spaced at 2.5 km, 6 km, 1 km, 8 km, 150 m and 300 m, respectively, for (a) Changjiang, (b)Songhua, (c) Po, (d) Yellow, (e) Åmose, and (f) Vejle rivers. Please refer to **Figure A1** for a detailed map. Note that, Manning's coefficient used for calculation of conveyance for each cross-section is randomly generated between 0.015 and 0.05.

However, there are four more parameters (i.e. $\alpha$, $\beta$, $\gamma$, $\delta$) for each cross-section to be estimated. Due to the linear nature of logarithmic pairs of $(A, d)$ and $(K, d)$, a linear relationship can be derived between the $A \sim d$ and $K \sim d$ curves for each individual cross-section. The mathematical derivations are given in Appendix A (Eqs. (A1) – (A4)). Unfortunately, the linear

relationships are varying with cross-sections, and therefore many coefficients (intercepts and slopes of the linear function as shown in Eqs. (A2) and (A4)) have to be determined. By fitting a linear function to the cross-section parameter pairs ($\alpha \sim \delta$ and $\beta \sim \gamma$) derived from observed data at the reach scale as shown in Figure 2, the flow area and conveyance curves for all cross-sections can be connected by:

$$\alpha = p_1 + p_2 \gamma, \tag{11}$$

$$\beta = p_3 + p_4 \delta. \tag{12}$$

It should be noted that the linear relationships (i.e. Eq. (11) and Eq. (12)) are only valid at river reach scale instead of individual cross-sections. In this way, we can simplify the hydraulic inverse problem by tying there parameters together, i.e. halving the number of fitting parameters. Interestingly, $p_1$, $p_2$, $p_3$, and $p_4$ are nearly constant independent of rivers although marginal deviations exist (Figure A4). As shown in Figure 2, when pooling cross-sections of all rivers together, a clear linear trend shows up for both $\alpha \sim \gamma$ and $\beta \sim \delta$. This indicates that parameters $p_2$ and $p_4$ should vary in a very narrow range around 1.0 for all rivers; And parameters $p_1$ and $p_3$ should be allowed to be slightly varying around - 1.4 and - 0.7 to adapt to individual rivers. Thus, there are two spatially varying parameters (i.e. $\gamma$, $\delta$) and four uniform parameters (i.e. $p_1 \sim p_4$) in addition to the bed datum $Z_0$, from which the bed slope $S_o$ is calculated, which have to be constrained in order to solve $Q$ and $d$. Therefore, a new parameterization of a river model can be written as:

$$\log_{10}\big(K(x,t)\big) = \gamma(x) + \delta(x) \, \log_{10}\big(d(x,t)\big), \tag{13}$$

$$\log_{10}\big(A(x,t)\big) = \big(p_1 + p_2 \, \gamma(x)\big) + \big(p_3 + p_4 \, \delta(x)\big) \log_{10}\big(d(x,t)\big), \tag{14}$$

with $p_1$, $p_2$, $p_3$, and $p_4$, close to - 1.4, - 0.7, 1.0 and 1.0, respectively.

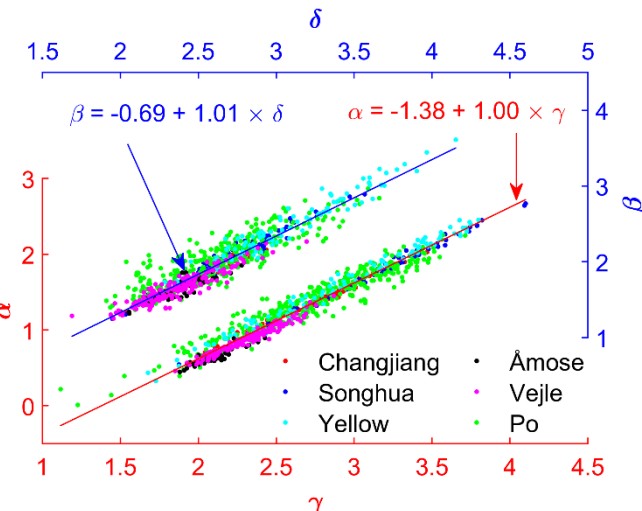

**Figure 2.** Linear relationship between **$\alpha/\beta$** and **$\gamma/\delta$** . Each dot represents one cross-section of a certain river. Dots of the same color are from the same river. Manning's coefficient for each cross-section is randomly generated between 0.015 and

0.05. Note that, the best-fit line for each river is slightly different. The relationship using a uniform Manning's coefficient of 0.03 is also given in **Figure A3**. Individual fitting lines are shown in **Figure A4**.

## 2.3 Parameter calibration

Hydraulic parameter calibration is essentially an inverse problem that is often solved using the least squares approach. Considering the large number of parameters ($p_1$, $p_2$, $p_3$, and $p_4$, and spatially varying $Z_0$, $\gamma$ and $\delta$), regularization is used to stabilize the ill-posed problem (Pereverzyev et al., 2006; Schmidt, 2005). In this work, the Tikhonov type regularization is applied and the objective function is formulated following (Aster et al., 2018):

$$\emptyset(\mathbf{X}) = \lambda\ misfit + (1 - \lambda)\ reg, \tag{15}$$

$$misfit = w \sum \frac{1}{N_h}\left(\frac{h_s - h_o}{\sigma_h}\right)^2 + (1 - w) \sum \frac{1}{N_b}\left(\frac{b_s - b_o}{\sigma_b}\right)^2, \tag{16}$$

$$reg = \lambda_\gamma \sum \frac{1}{N}\left(\frac{L\,\gamma}{\sigma_\gamma}\right)^2 + \lambda_\delta \sum \frac{1}{N}\left(\frac{L\,\delta}{\sigma_\delta}\right)^2 + \lambda_{p1} \sum \left(\frac{p_1 + 1.4}{\sigma_{p1}}\right)^2 + \lambda_{p2} \sum \left(\frac{p_2 - 1}{\sigma_{p2}}\right)^2 + \lambda_{p3} \sum \left(\frac{p_3 + 0.7}{\sigma_{p3}}\right)^2$$

$$+ \lambda_{p4} \sum \left(\frac{p_4 - 1}{\sigma_{p4}}\right)^2 + \left(1 - \lambda_\gamma - \lambda_\delta - \lambda_{p1} - \lambda_{p2} - \lambda_{p3} - \lambda_{p4}\right) \sum \frac{1}{N}\left(\frac{L\,z}{\sigma_z}\right)^2, \tag{17}$$

where $\mathbf{X}$ is vector containing the parameters $\gamma$, $\boldsymbol{\delta}$, $p_1$, $p_2$, $p_3$, $p_4$, and $z$; $\lambda$ is a weighting factor balancing the regularization and data fitting error; $w$ is a weighting factor balancing the fitness of water level and width; $\boldsymbol{h_s}$, $\boldsymbol{h_o}$, $N_h$, $\sigma_h$ are simulated water level, observed water level, number of water levels, and the uncertainty of observed water level; $\boldsymbol{b_s}$, $\boldsymbol{b_o}$, $N_b$, $\sigma_b$ are simulated width (calculated as the derivative of flow area with respect to depth in the model), observed width, number of widths, and the uncertainty of observed width; $\lambda_\gamma$, $\lambda_\delta$, $\lambda_{p1}$, $\lambda_{p2}$, $\lambda_{p3}$, $\lambda_{p4}$ are regularization parameters; $\sigma_\gamma$, $\sigma_\delta$, $\sigma_{p1}$, $\sigma_{p2}$, $\sigma_{p3}$, $\sigma_{p4}$ $and$ $\sigma_z$ are the a priori standard deviations indicating how uncertain the parameters are a priori; $N$ is the number of cross-sections; and $\boldsymbol{L}$ is the first-order regularization roughening matrix, which is a finite-difference approximation to the first derivative of the model:

$$\boldsymbol{L} = \begin{bmatrix} 1 & -1 & & & & \\ & 1 & -1 & & & \\ & & \ddots & \ddots & & \\ & & & 1 & -1 & \\ & & & & 1 & -1 \end{bmatrix}.$$

In this case study, the weighting factor $\lambda$ is set as 0.8 based on a trial-and-error method. The weighting factor $w$ is set to 0.5, i.e. water level and river width observations are equally important. The uncertainties of water level and width are 0.5 m and 99 m according to (Jiang et al., 2017) and (Yang et al., 2020), respectively. The a priori standard deviations of $\sigma_\gamma$ and $\sigma_\delta$ are 0.7 and 0.4, respectively, which are similar for relatively large rivers (see $\gamma$ and $\delta$ distributions in Figure S2). The a priori standard deviations of $p_1$, $p_2$, $p_3$, $p_4$ are chosen as 0.02, 0.01, 0.02, and 0.01, respectively, given that those parameters vary slightly (see Figure A3 and Figure A4). The datum $Z_0$ is the sum of parameter $z$ and a constant value which is estimated from

the average water level subtracting the depth of 5 m. The a priori standard deviation of $z$ is 0.5 m. The regularization parameters, i.e. $\lambda_\gamma$, $\lambda_\delta$, $\lambda_{p1}$, $\lambda_{p2}$, $\lambda_{p3}$, $\lambda_{p4}$ are empirically set as 0.1, 0.1, 0.15, 0.15, 0.15, 0.15, respectively, to achieve appropriate smoothness.

We iteratively optimize the objective function (Eq. (15)) with the Levenberg-Marquardt (LM) algorithm (Marquardt, 1963) combined with Broyden's rank-one update to approximate the Jacobian (Broyden, 1965; Madsen et al., 2004). We use
an implementation of the method provided by the Immoptibox toolbox (Nielsen and Völcker, 2010). Given that LM finds a local minimum and cannot guarantee the global minimum, an ensemble of 10 calibrations is carried out with different initial guesses to avoid convergence to a local minimum. For each calibration, the number of model runs is around 200. The time consumed for this optimization is a few hours (1 ~ 4 hours). The calibrations were conducted on a windows server 2016 (Inter® Xeon® Gold 6154 CPU @ 3 GHz, 2993 Mhz) using 4 cores.

The calibration is implemented in Matlab. C# scripts are used to modify and dump MIKE Hydro River parameters and simulation results. The power-law relationships are an integral part of the MIKE Hydro River model. Specifically, for each iteration of the optimization, the updated parameters by LM algorithm as well as the calculated flow area and conveyance relationships are passed to a C# script that updates the setup of the MIKE Hydro River model. Then the model is executed, and the results are passed on to Matlab. Essentially, by optimizing Eq. (15) using satellite derived observations of WSE and
river width, we calibrate the two curves, i.e., the relationships between flow area / conveyance and depth as described by equations (13) and (14) for each cross-section along the reach.

## 3 Case study

To test whether this approach is able to reproduce realistic flow area and conveyance curves as well as WSE using remote sensing data, we use the Songhua River as a test site. Below are the descriptions of test site, data sets, model setup and
calibration procedures.

### 3.1 Study site

Songhua River is the longest tributary of the Amur (or Heilong Jiang), and one of the largest rivers in the world. It allows testing the approach using satellite data sets, such as altimetry and imagery, which will be available simultaneously from the future SWOT mission (Biancamaria et al., 2016).

The river has two sources, i.e. Nenjiang and Second Songhua rivers, originating from the Greater Khingan Range in the north and the Paektu Moutain in the south, respectively, and drains an area of 556 800 km$^2$. At Sanchahe, two tributaries merge to form the Songhua River. It runs 840 km northeastward before draining into the Amur River (Songliao River Conservancy Commission, 2004, 2015). In this study, we focus on the middle reach of the Songhua River, between Harbin and Jiamusi (Figure 3). The reasons why we selected this reach are twofold: firstly, it is wide enough (700 m on average) to

have high-quality altimetry data as shown in previous study (Jiang et al., 2017) and secondly, we have access to in-situ data

of several hydrometric stations across this region. This reach covers an area of 138 500 km² and stretches 433 km long. The

elevation difference of this reach is about 45 m, resulting in an average slope of 0.1 m km⁻¹. The first 222 km flows through

hilly terrain with a gentle slope of 0.05 m km⁻¹ while the downstream reach is narrower and deeper. The mean discharge at

the downstream end is about 1175 m³ s⁻¹. The river is frozen in winter and reaches its maximum flow in summer.

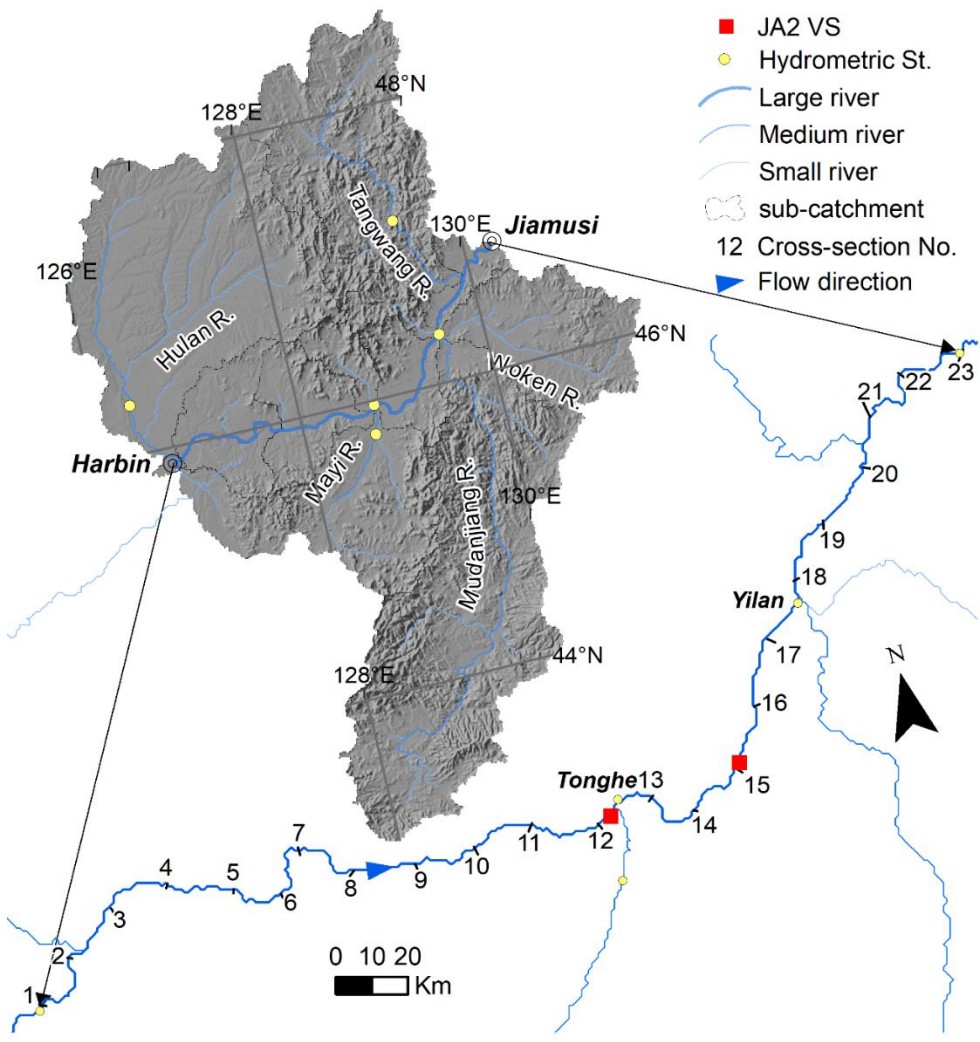

**Figure 3.** Overview of the study area. The studied reach is 433 km long between Harbin and Jiamusi. There are five major
tributaries recharging the main river. There are 23 cross-sections evenly distributed along this reach as shown in the lower
map.

**3.2 Data sets and model setup**

A 1D river model is built using the MIKE Hydro River software (DHI, 2017). The first step is to define the river network, cross-sections, and boundary conditions. The river network is set up using the center line of the reach, while 23 cross-sections are equally distributed along the 433 km reach as in Jiang et al. (2019). The daily discharge at Harbin hydrometric station is used as the upstream boundary while a uniform flow depth rating curve is set as downstream boundary. Inflows of three tributaries are from gauging records while remaining tributary inflows are simulations from a hydrological model

(Jiang et al., 2019). The only available in-situ surveyed cross-sections are from the late 1990s. These "real" area curves are only used to validate the calibration results.

WSE and river width derived from CryoSat-2 altimetry and Landsat imagery are used as observations in the calibration. CryoSat-2 altimetry is distinctive due to several features, most importantly the orbit configuration. Specifically, CryoSat-2 with its drifting ground track pattern results in an entirely different sampling pattern. The small inter-track spacing of 7.5 km

enables dense sampling of rivers and thus provides longitudinal water level profiles. Although these profiles are not snapshots of river level at a given time, they are still useful for resolving local hydraulic characteristics (Jiang et al., 2019; Schneider et al., 2018). CryoSat-2 observations are the same as those used in Jiang et al. (2019), covering the period of 2010-2014. Widths are extracted using the RivWidthCloud algorithm in Google Earth Engine (Yang et al., 2020). We used Landsat 5 and Landsat 8 images and selected images avoiding cloud cover and obtain an even distribution in time.

Specifically, if the river is cloud-free in a given image, it is selected regardless of the cloudiness of other parts. Images collected from December to early April are excluded. In total, 37 Landsat 5 images and 15 Landsat 8 images are used and provided 10022 individual width observations. The temporal and spatial distribution of WSE and widths observations is shown in Figure B1.

For the purpose of validation, gauging records of water level and discharge at Tonghe and Yilan (Figure 3) are

collected. Moreover, WSE datasets derived from Jason-2 at two virtual stations (Figure 3) are also included for extensive validation. Because the river is completely frozen, altimetry does not provide realistic WSE observations during the winter. Therefore, we only consider the ice-free period (April to October) in this study. To compare results with the previously published calibration approach (e.g. simultaneous calibration of roughness and cross-section shape parameters), we also extract model simulations from our previous work (Jiang et al., 2019). Specifically, water level simulations from model

calibration S1 (refer to Jiang et al. (2019)) are used for a fair comparison given that both calibrations use the same amount of CryoSat-2 WSE data.

**3.3 Calibration scenarios**

To test the capability of different data sets to constrain model parameters, three basic scenarios are used based on the type of data sets. That is, calibration #1 uses altimetry derived WSE only; calibration #2 uses imagery derived width only; and

calibration #3 uses both WSE and width. Given that width observations are of very high spatial resolution (30 m interval), three scenarios of width observations are also designed (Table 1). Specifically, width is sampled at coarse spatial resolution

by randomly selecting one observation for each 2 or 5 km reach regardless of the timing. Given that only 261 observations of WSE are available, no further exploration of the effect of WSE data is performed. Therefore, in total, we test 7 scenarios of observations to calibrate the model (Table 1).


Table 1. Details of the calibration scenarios with different data sets.

| Scenario | Description | Num. of WSE | Num. of width |
|---|---|---|---|
| Calibration #1 | Calibration with WSE observations only | 261 | 0 |
| Calibration #2a | Calibration with one width per 5 km | 0 | 88 |
| Calibration #2b | Calibration with one width per 2 km | 0 | 219 |
| Calibration #2 | Calibration with width observations only | 0 | 10022 |
| Calibration #3a | Calibration with WSE and one width per 5 km | 261 | 88 |
| Calibration #3b | Calibration with WSE and one width per 2 km | 261 | 219 |
| Calibration #3 | Calibration with WSE and width observations | 261 | 10022 |

## 4 Results

Results prove that it is possible to calibrate spatially varying area-depth curves using solely satellite data sets. Figure 4 depicts the calibrated area-depth curves at 23 cross-sections under the three scenarios. Two metrics are used to evaluate the performance alongside the plots. RMSE describes the error of the calibrated $\log Area$ vs $\log Depth$ relationship, and

coverage is defined by the percentage of real data that fall within the confidence interval. Compared to the curves derived from surveyed cross-sections, the calibrated ones are reasonably close at most locations. Most of the largest errors occur at cross-sections 4-6, where the Dadingzishan reservoir (chainage 20 - 90 km) is located and not modelled. Interestingly, both WSE alone and river width alone are able to constrain the model to a certain degree (Figure 4). However, calibration #1 (WSE only) has slightly larger spread especially for small depths. The average RMSE and coverage are 0.42 and 16%.

Calibration #2 (width only) tends to overestimate flow area, which is significant for downstream cross-sections. The corresponding average RMSE and coverage are 0.34 and 8%. Calibration #3 (both WSE and width) shows the best match (smaller RMSE and larger coverage) with the observed cross-sections (Figure 4). Moreover, very dense observations of width (#2 and #3) do not improve the calibration results compared to less dense ones (#2a and #2b, #3a and #3b) although calibrations #2 and #3 result in narrower confidence intervals (Figure B2 and Figure B3).

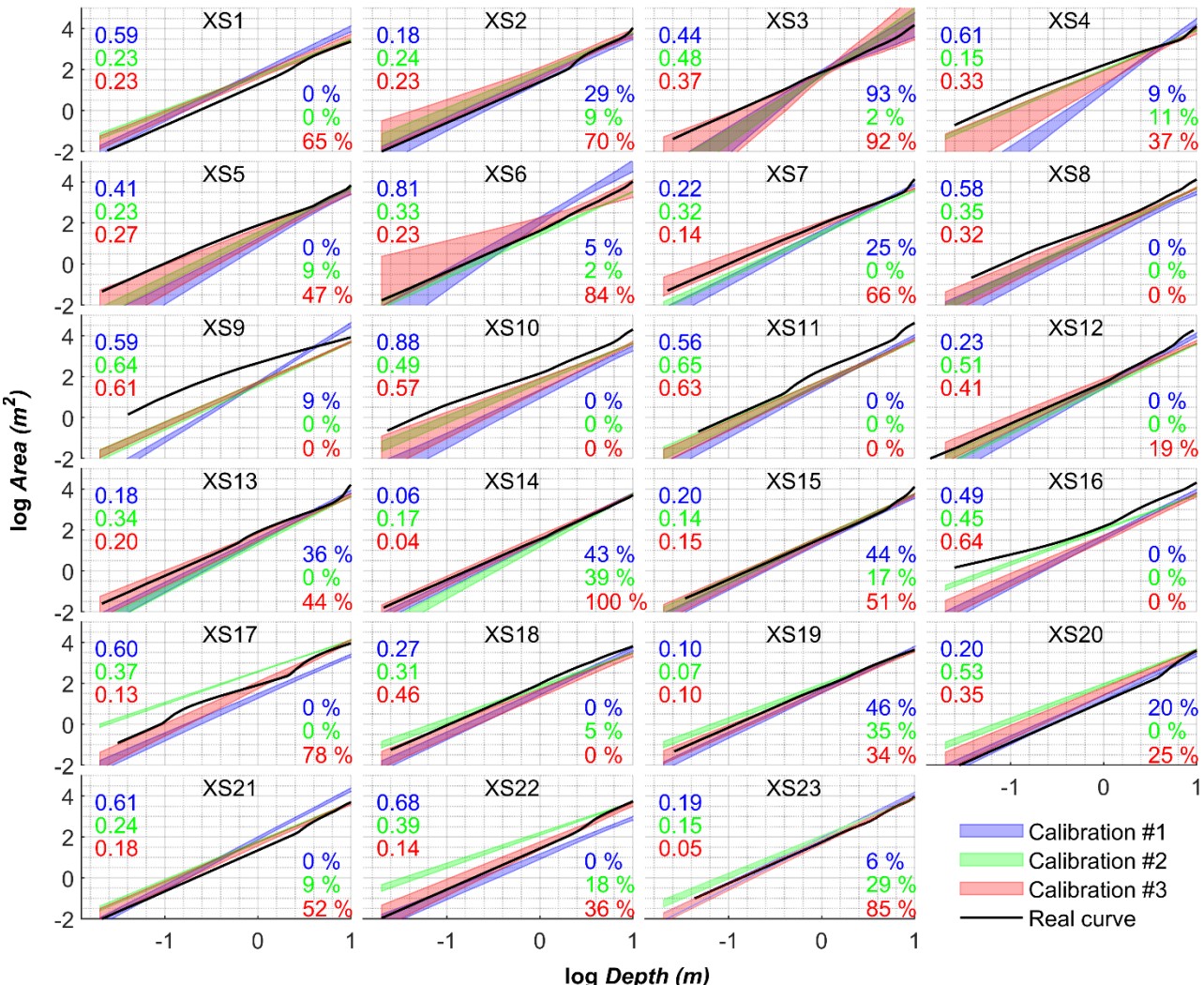

**Figure 4.** Calibrated area-depth curves at 23 cross-sections (number is given in each plot). Three scenarios are shown, i.e. calibration with water surface elevation data only (calibration #1), river width only (calibration #2), and both water surface elevation and width (calibration #3), respectively. Please refer to Table 1 for more information. The color band represents the mean ± standard deviation based on an ensemble of 10 calibrations. RMSE and coverage (percentage of real data falling into the calibrated interval) of calibrated curves against real data are reported on the left and right sides of each plot, respectively. Font color is consistent with the curve color.

Figure 5 Shows the performance of each calibration scenario in terms of accuracy of simulated water level. Similarly, models calibrated with either WSE (calibration #1) or width (calibration #2) can reproduce WSE with similar RMSE at two gauging stations. However, calibration #2 shows larger RMSEs and wider ranges than calibration #1, especially at the Yilan station. In contrast, calibration #3 is more stable, resulting in smaller RMSEs and narrower ranges. This is in line with the well calibrated area-depth curves at cross-sections (XS12, XS13, XS17, XS18) nearby the two gauging stations (refer to Figure 3 for the locations). Regarding the scenarios using width observations, the RMSE values of calibrations #2a and #2b

are very spread (i.e. a wide range), indicating that models are not well-constrained. This is evidenced by the poorly calibrated area-depth curves (e.g. wider color bands of XS17 and XS18) shown in Figure B2.

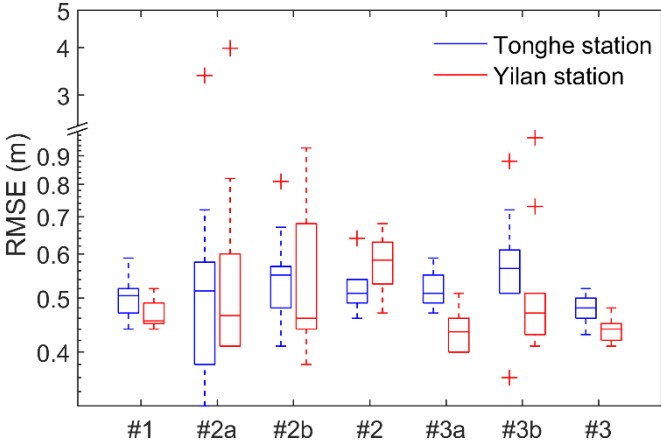


**Figure 5.** Boxplots of evaluations of simulated water level against in-situ gauging records at two gauging stations. Calibration scenarios indicated on the x-axis are referred to **Table 1**. Note that, the y-axis is in log scale. Note that, the statistics for each scenario are computed from the 10 calibration runs with different starting points.

The calibrated model can reproduce the WSE reasonably well when compared with independent data sets. Figure 6
shows simulated WSE using calibrated curves shown in Figure 4. Overall, the accuracy of simulation is acceptable. The RMSE is about 50 cm and 44 cm at Tonghe and Yilan stations, respectively. The accuracy is comparable to what was achieved using a different approach, which simultaneously calibrates cross-section shape parameters and roughness (Jiang et al., 2019). A careful comparison indicates that the simulations are slightly better than those reported in Jiang et al. (2019) for low WSE (Figure 6). Compared to Yilan, Tonghe shows slightly higher RMSE (Figure 6) due to the underestimation of the
extremely high WSE in 2013, although the simulated discharge matches in-situ observations well (Figure C1). This can be well explained by the calibrated curves. The curves at two neighboring cross-sections (XS12 and XS13) show deviations from the curves derived from surveyed cross-sections beyond bankfull depth (upward curves as shown in Figure 4). Evaluation at the two virtual stations also shows good agreements. However, the model simulation is better than Jason-2 observations except during the 2013 flood when compared to the hydrograph of an adjacent gauging station, i.e. Tonghe
station (Figure 6).

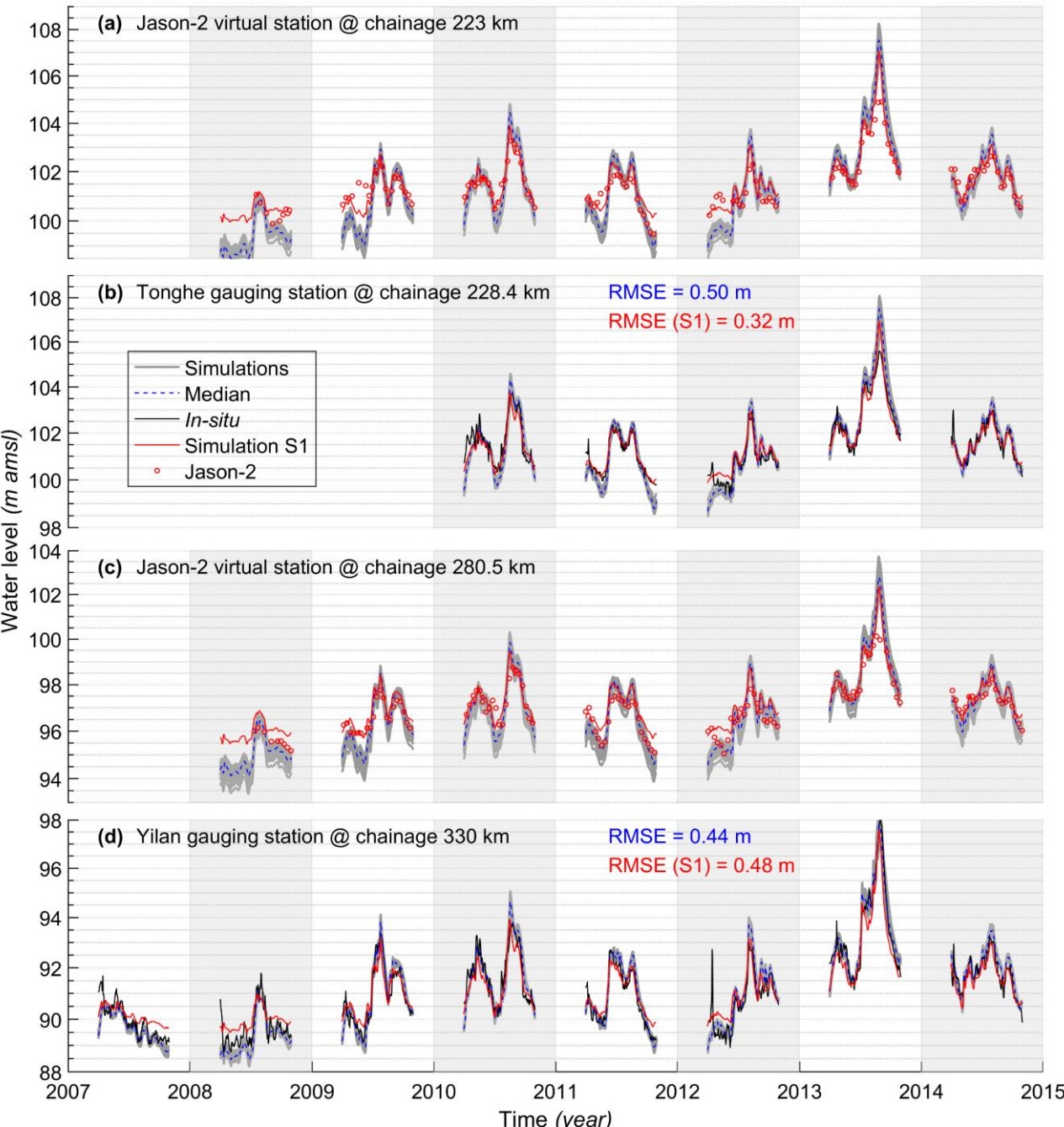

**Figure 6.** Validation of simulated water level (non-frozen periods) at four stations. (a) and (c) are water levels at two virtual stations, i.e. data derived from Jason-2 altimetry. (b) and (d) are from two stream gauging stations. Simulation from previous model calibrated using different strategy (i.e. simultaneous calibration of roughness and cross-section shape parameters; simulation S1 in Jiang et al. 2019) is also shown for comparison. Note, in each plot, results of the median and individual simulations of an ensemble of 30 calibrations (#1, #2, and #3) are shown.

## 5 Discussion

### 5.1 The value of altimetry and imagery in model calibration

Satellite altimetry data and imagery data have been increasingly used to calibrate hydrologic and hydrodynamic river models (Domeneghetti et al., 2014; Jiang et al., 2019; Liu et al., 2015; Michailovsky et al., 2012; Milzow et al., 2011; Sun et al., 2010). However, joint use of imagery and altimetry for hydrodynamic modeling is not common practice. For our case study, models calibrated with either river width only or WSE only show similar performance in terms of RMSE of WSE at two gauging stations (Figure 5). However, both cases have problems to fully constrain parameters and suffer from model ambiguity, which means parameters cannot be well determined. A direct consequence is that model simulations of either the WSE or river width are not physically meaningful (Figure C2). Because both cases can achieve a reasonable area-depth relationship by making a trade-off between datum and WSE or river width. For example, calibration #1 (WSE only) shows reasonable simulation of WSE (Figure 5), but the simulation of width is not meaningful (Figure 5 and Figure C2). Therefore, both WSE and river width are needed to better constrain model parameters.

Nevertheless, river width and WSE may play different roles in constraining parameters for different rivers depending on the channel shape. If a channel is embanked, for instance, model parameters may not be sensitive to the small changes of river width. This issue certainly needs further investigation. Obviously, observations of river width are easier to obtain and have higher frequency and larger coverage than altimetry-derived WSE (usually the frequency is lower than 10 days). That is, this approach can be applied in many rivers where both altimetry data and imagery are available given reliable discharge at the upstream boundary. This raises a question: Can area and conveyance curves be estimated using short-repeat altimetry missions, such as Jason or Sentinel-3? Our previous study (Jiang et al., 2019) shows that spatial sampling density is more important than temporal frequency in the context of hydraulic inversion and that Jason series alone are not able to constrain the spatially distributed parameters. The trade-off between spatial and temporal sampling density in inland radar altimetry merits further investigation. Moreover, rapid advances in drone technology also provide WSE and width for small rivers (Bandini et al., 2020). Therefore, this approach is also applicable to rivers where satellite altimetry data are not available. Moreover, further comprehensive investigation of the impact of width observations (i.e. image spatial resolution and temporal distribution, accuracy, etc.) is needed to draw solid conclusions. Ongoing research will employ simultaneous observations of river width and WSE from SWOT for river hydrodynamic modeling.

### 5.2 Implications for hydrodynamic modeling in ungauged catchments

Lack of river channel bathymetry data restricts application of hydrodynamic modeling to data-scarce river basins. Most continental- or global-scale hydrologic models are coupled with simple routing schemes for simulating surface water transport in the major rivers of the world (Yamazaki et al., 2011). However, at basin scale, without explicit representation of channel geometry, resolving water level dynamic is impossible. Coupling hydrologic models with hydrodynamic river models would better describe the flow dynamics (water depth, water level, discharge, etc.). The new parameterization

proposed in this paper can also be used with simulated discharge (from a hydrologic model) instead of observed discharge.
In this way, water levels along the river channel could be resolved. We performed a preliminary investigation into the effect of simulated discharge errors on inverted area and conveyance curves. Specifically, for the upstream boundary, modelled discharge from a regional rainfall-runoff model is used instead of in-situ discharge (Figure C3). With this setup, the calibrated 1-D hydrodynamic model can reproduce WSE reasonably well (~0.9 m, see Figure C3). The accuracy is comparable to previous studies, such as Domeneghetti et al. (2014) although surveyed cross-sections were used in those studies. This finding demonstrates that this approach has great potential to be applied in ungauged river basins. This is in line with the statement by Liu et al. (2015) that in-situ discharge data may not be necessary for successful hydrologic model calibration. However, more research is needed to incorporate the proposed parameterization into fully coupled hydrologic-hydrodynamic models for ungauged basins.

### 5.3 Known issues and limitations

The power-laws of flow area / conveyance and flow depth and the corresponding linearity approximation are confirmed in 6 rivers. One may argue that the relationship may not be globally applicable due to the limited number of rivers to validate the relationships. We cannot rebut this argument without collecting a large sample of surveyed cross-sections, which is difficult because of data access problems. However, the rivers we used are of diverse sizes (width ranging from a few meters to kilometers) and flow characteristics, and are from different climate zones (Arctic, Mediterranean, and Asian temperate climates). Therefore, we believe that the relationship holds globally, and we call for extensive validation using other rivers. Regarding the linear relationship between the flow area and conveyance curves for each cross-section, it is understandable intuitively given that both flow area and conveyance are linearly related to the same variable, i.e. flow depth. At river reach scale, the strong linear relationships between $\alpha$ and $\gamma$, $\beta$ and $\delta$ are empirical.

As we mentioned, this study only focuses on the main channel and does not account for overbank flow. In the presence of significant floodplains, the linearity of the curve may fail at bankfull depth as seen in Figure 1 and Figure 4. Consequently, as seen in Figure 6, the model over-estimates extreme flood peak (year 2013). Similarly, one curve may not be able to describe anastomosing rivers that consist of compound channels. To solve this problem, a second curve is needed to describe the overbank flow as suggested by Garbrecht (1990). On the other hand, instead of calibrating the second curve, real data (such as high resolution DEMs, or ICESat-2, etc.) of the non-inundated portion can be used to parameterize the curves instead or apply 1D-2D modeling in the case of significant floodplains. Moreover, this approach assumes that the established curves are time invariable, which is not applicable to rivers with significant bedform changes.

In summary, this approach opens up a range of possibilities to simulate and predict flow dynamics in data scarce regions. In addition to simulating WSE as illustrated in previous sections, discharge retrieval is also possible once the slope is known based on established conveyance curves. The future SWOT mission will deliver WSE and slope simultaneously, which can support discharge retrieval using this approach.

# 6 Conclusions

Directly calibrating roughness and cross-section geometry of river models is still challenging. In this paper, we propose an alternative approach to calibrate 1-D hydrodynamic river models using altimetry and imagery observations. The workflow is based on the power-law relationships between flow area / conveyance and flow depth, which goes back to Chow (1959). In

this study, we discovered that the two curves are very well correlated, and applicable for a wide range of rivers. The novelty of this study is that the flow area and conveyance can be inverted directly using spatially distributed observations of WSE and river width given the boundary conditions. In this way, the roughness and channel geometry do not have to be explicitly known to determine the WSE.

Our case study demonstrates that the curves can be estimated using solely remote sensing data, and the calibrated

hydrodynamic model can reproduce the WSE with high precision (ca. 40 - 50 cm). Our method performs comparably to existing ones which use conventional parameterization and calibration approaches. Further exploration indicates that our approach can be integrated into a hydrologic-hydrodynamic model for studying ungauged river basins.

Overall, this study provides an alternative method for hydrodynamic modeling, especially in regions without in-situ river cross-section data. Current satellite imagery (Landsat, Sentinel, Gaofen, etc.) and altimetry (CryoSat-2, AltiKa-DF) can

support this approach for relatively large rivers. This approach of parameterization and calibration may prove especially useful for poorly gauged rivers when high resolution data sets are available from the upcoming SWOT mission.

*Appendix A:* **Supplementary information on the relationships between flow area / conveyance and depth**

Take the linear relationships of (A, d) and (K, d), i.e. Eq. (8) and Eq. (9) as the start. Substituting $\log d$ in Eq. (9), we get

$$\log A = \alpha + \frac{\beta}{\delta}(\log K - \gamma). \tag{A1}$$

By rearranging Eq. (A1), we have

$$\alpha = \log A - \frac{\beta}{\delta}\log K + \frac{\beta}{\delta}\gamma. \tag{A2}$$

Further, we can write it as $\alpha = m + n\gamma$ with $m = \log A - \frac{\beta}{\delta}\log K$ and $n = \frac{\beta}{\delta}$.

Therefore, $\alpha$ is linearly related to $\gamma$ although the intercept ($m$) and slope ($n$) are not constant. That is, Eq. (A2) is valid at each individual cross-section.

Similarly, we divide Eq. (6) by Eq. (7), we can obtain

$$\frac{A}{K} = \frac{a}{c}d^{\beta-\delta}. \tag{A3}$$

Taking the logarithm and rearranging the equation, we have

$$\beta = \frac{\log A - \log K + \gamma - \alpha}{\log d} + \delta. \tag{A4}$$

Thus, $\beta$ can also be expressed as a linear function of $\delta$, but with varying intercept.

One should not confuse Eq. (A2) and Eq. (A4) with Eqs. (11) and (12). The formers are derived at individual cross-section, while the latter ones are derived by fitting linear functions to cross-section parameters at the reach scale.

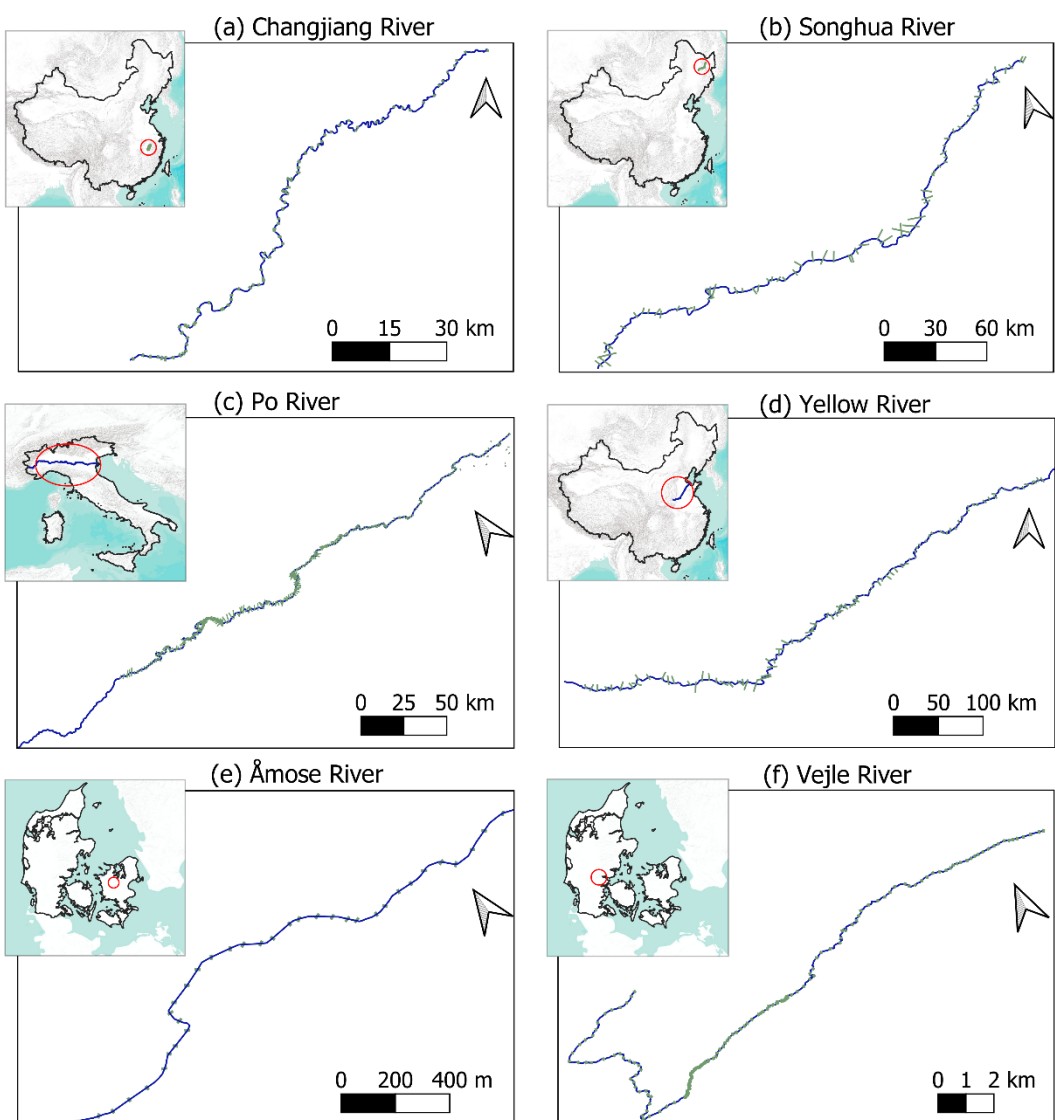

**Figure A1.** Location and river setting of six rivers. Grey short lines indicate cross-sections used to explore the hydraulic relationships.

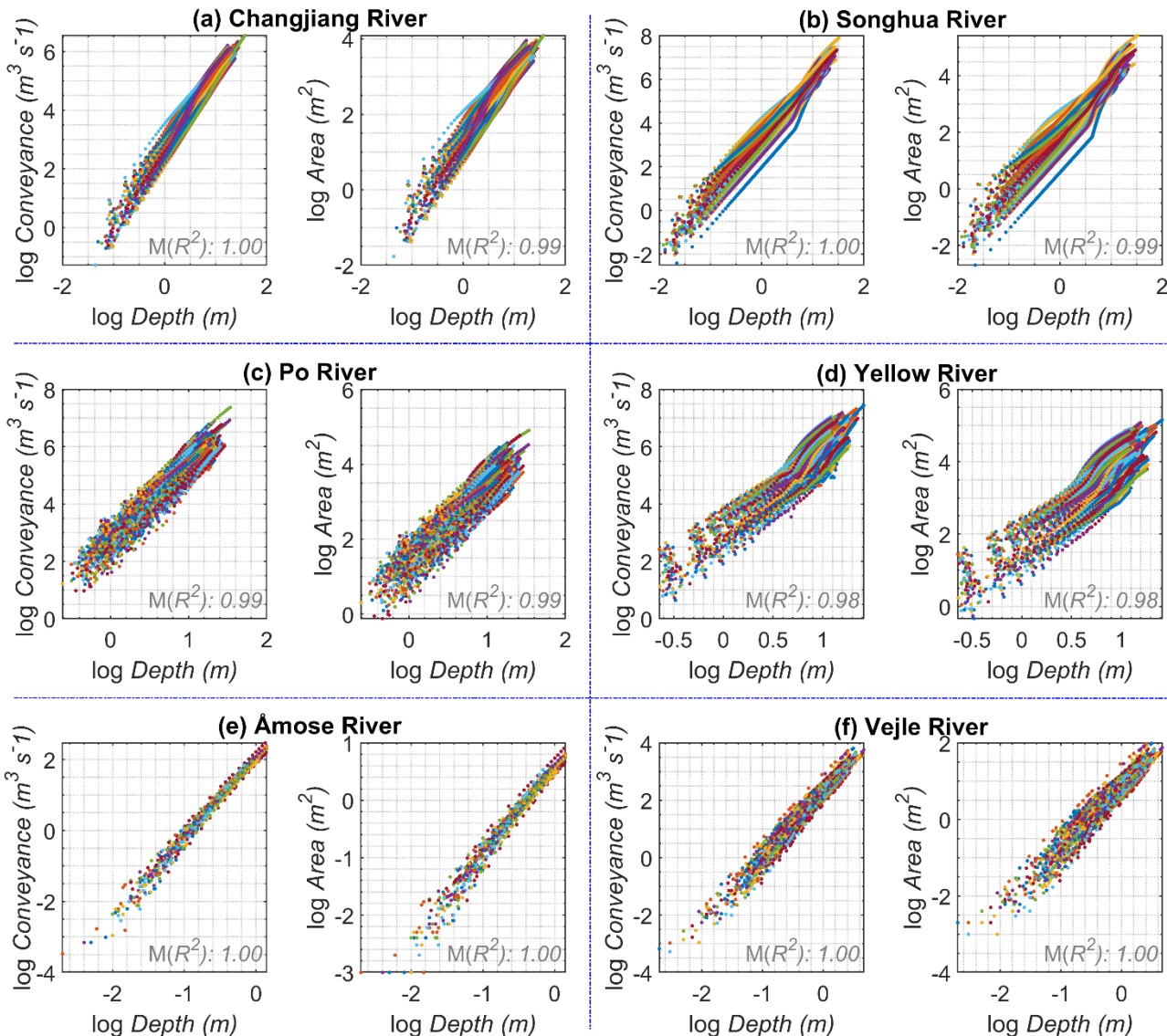

**Figure A2**. Relationship between logarithmic depth and logarithmic area and logarithmic conveyance. Similar to **Figure 1** but a uniform Manning's coefficient of 0.03 was used to calculate conveyance. This results in a stronger linear relationship. However, a uniform Manning's coefficient is not very realistic in natural rivers.

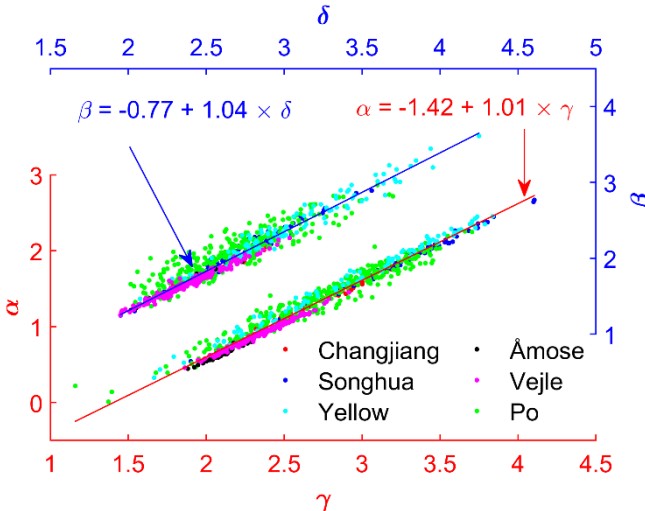

**Figure A3.** Linear relationships between gamma ~ alpha and delta ~ beta using data of all six rivers. Similar to **Figure 2**, but a uniform Manning's coefficient of 0.03 was used.

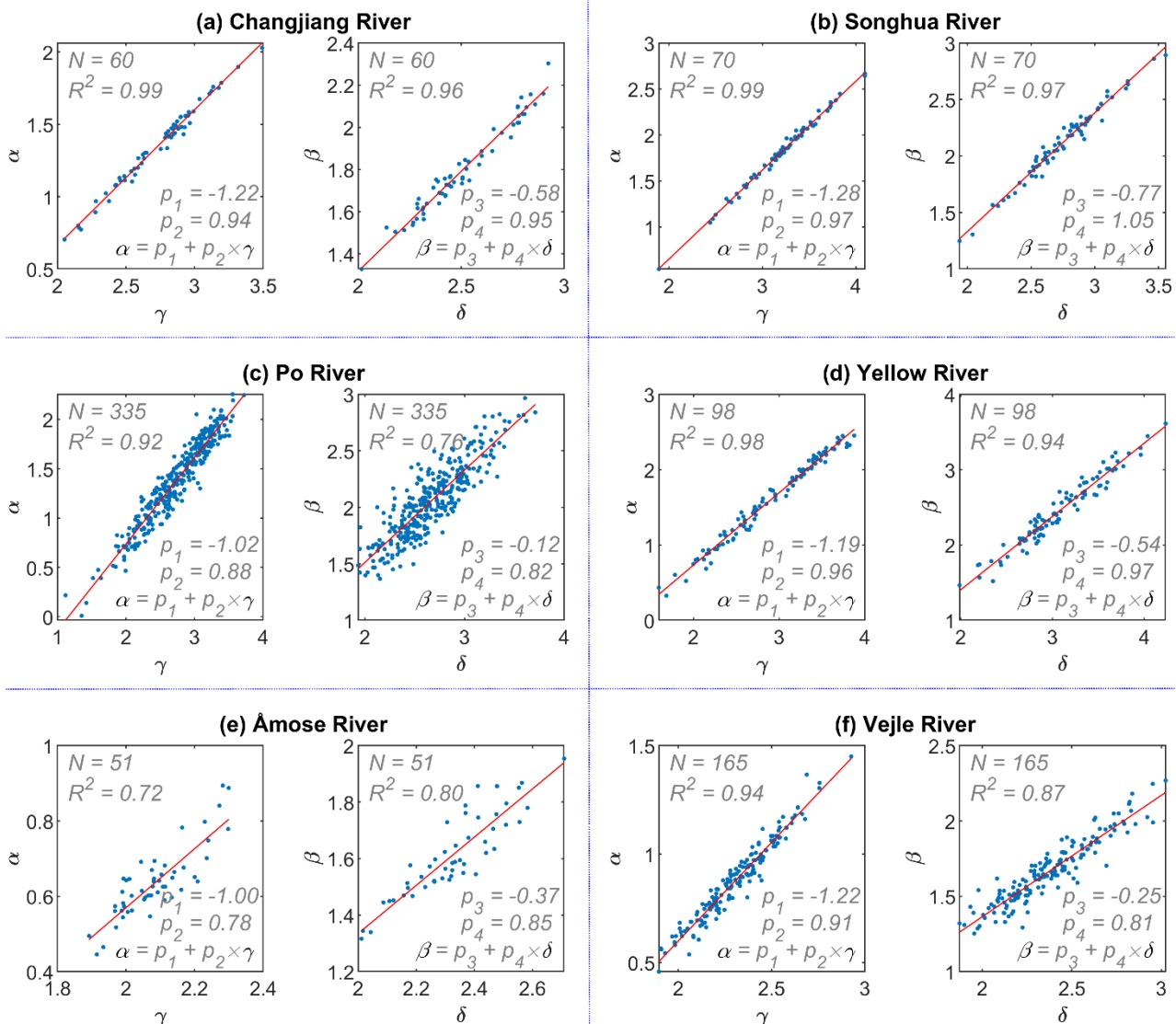

**Figure A4.** Linear relationships between gamma ~ alpha and delta ~ beta for six rivers. Randomly generated Manning's coefficient in range of 0.015 ~ 0.05 for each cross-section was used to calculate conveyance. The number of cross-sections, coefficient of determination, and regression coefficients are labelled in each plot.

*Appendix B:* **Supplementary information on the data sets and calibrated curves**

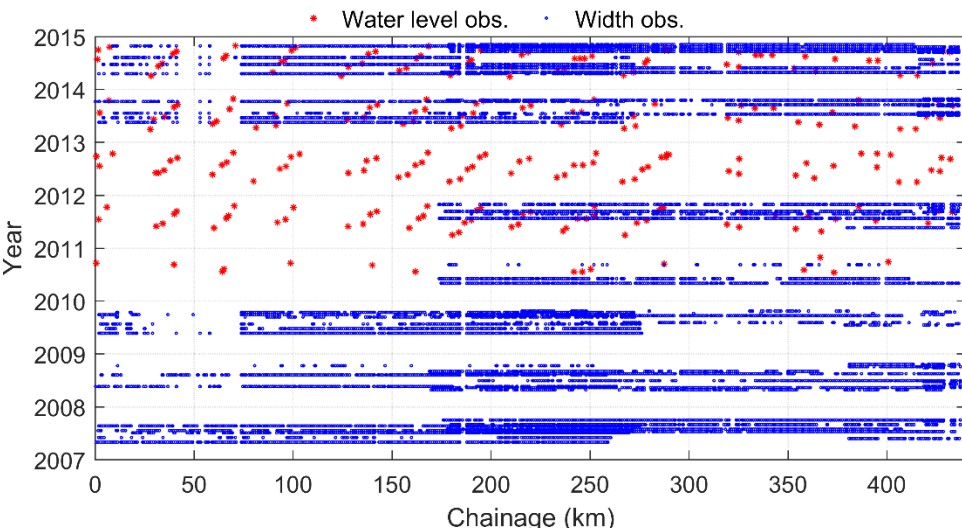

**Figure B1.** Temporal and spatial distribution of water levels and widths, which are used to calibrate the model. Given that the river is frozen in cold season, only data in warm seasons are used. Landsat-5/8 images with low cloud cover (visual checked in Google Earth Engine) are selected to generate river width.

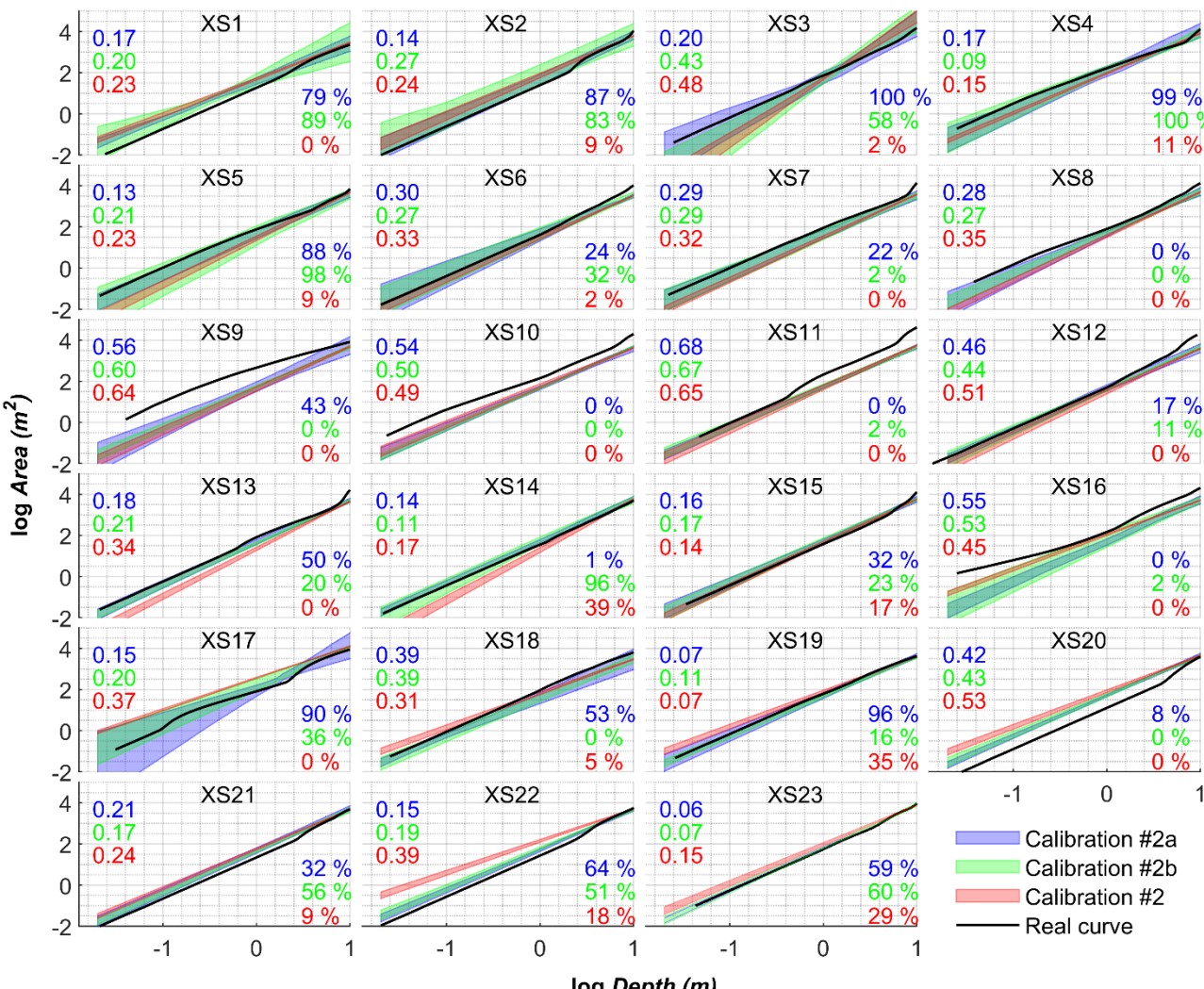

**Figure B2.** Calibrated curves using three scenarios of width observations only, i.e. one width observation per five-km reach (calibration #2a), one width observation per two-km reach (calibration #2b) and all available widths (calibration #2). The color band represents the mean ± standard deviation based on an ensemble of 10 calibrations. The number of cross-section is given in each plot. RMSE and coverage of calibrated curves against real data are reported on the left and right sides of each plot, respectively. Font color is consistent with the curve color. The average RMSE and coverage are 0.28, 0.30, 0.34 and 45%, 36%, 8% for calibration #2a, #2b, #2, respectively.

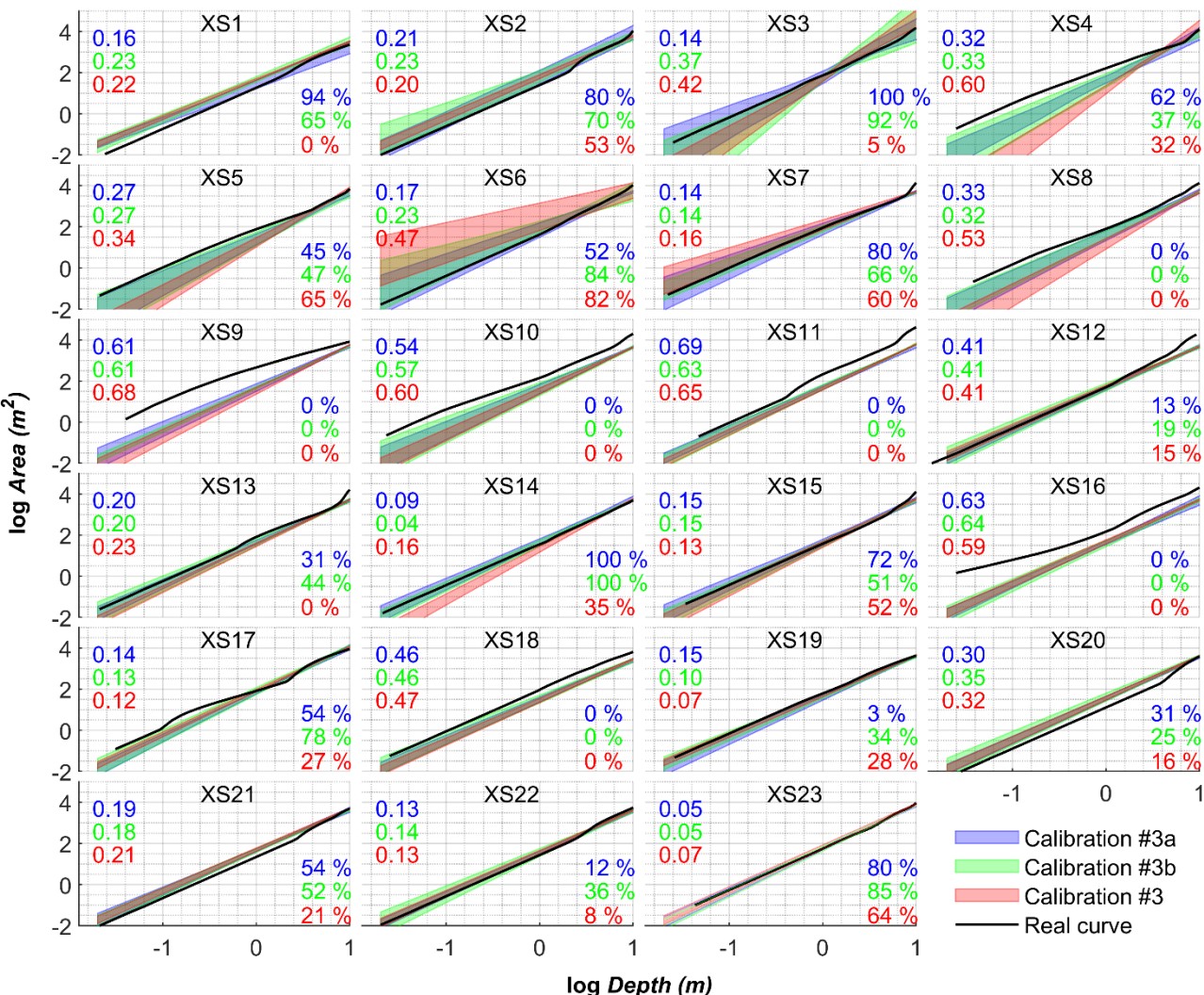

**Figure B3**. Calibrated curves using **three scenarios of width** observations and water level, i.e. one width observation per five-km reach and water levels (calibration #3a), one width observation per two-km reach and water levels (calibration #3b) and all available widths and water levels (calibration #3). The color band represents the mean ± standard deviation based on an ensemble of 10 calibrations. The number of cross-section is given in each plot. RMSE and coverage (percentage of real data falling into the calibrated interval) of calibrated curves against real data are reported on the left and right sides of each plot, respectively. Font color is consistent with the curve color. The average RMSE and coverage are 0.28, 0.29, 0.34 and 41%, 42%, 24% for calibration #3a, #3b, #3, respectively.

 *Appendix C:* **Supplementary information on the simulation results**

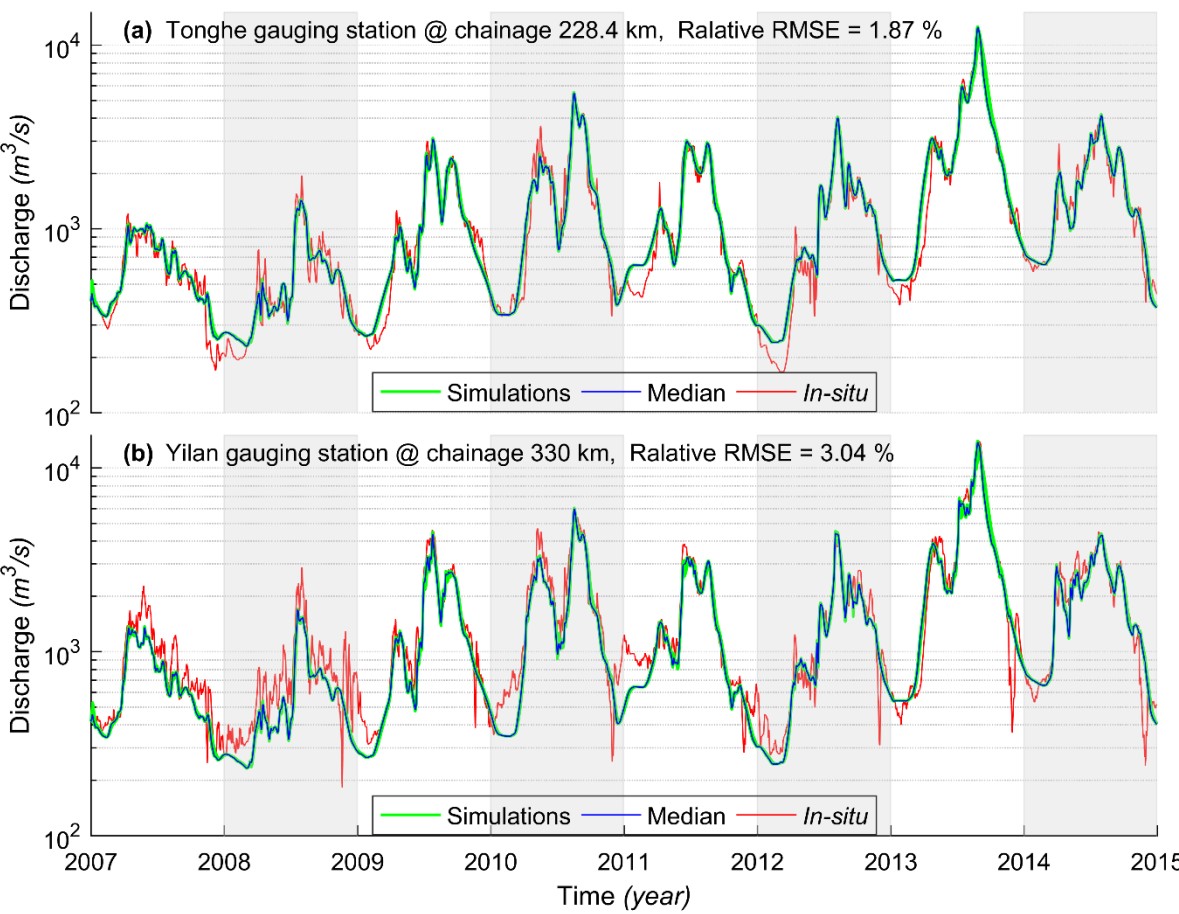

**Figure C1**. Validation of simulated discharge at two gauging stations. Note, in each plot, results of the median and individual simulations of an ensemble of 30 calibrations (i.e. calibration #1, #2, and #3) are shown.

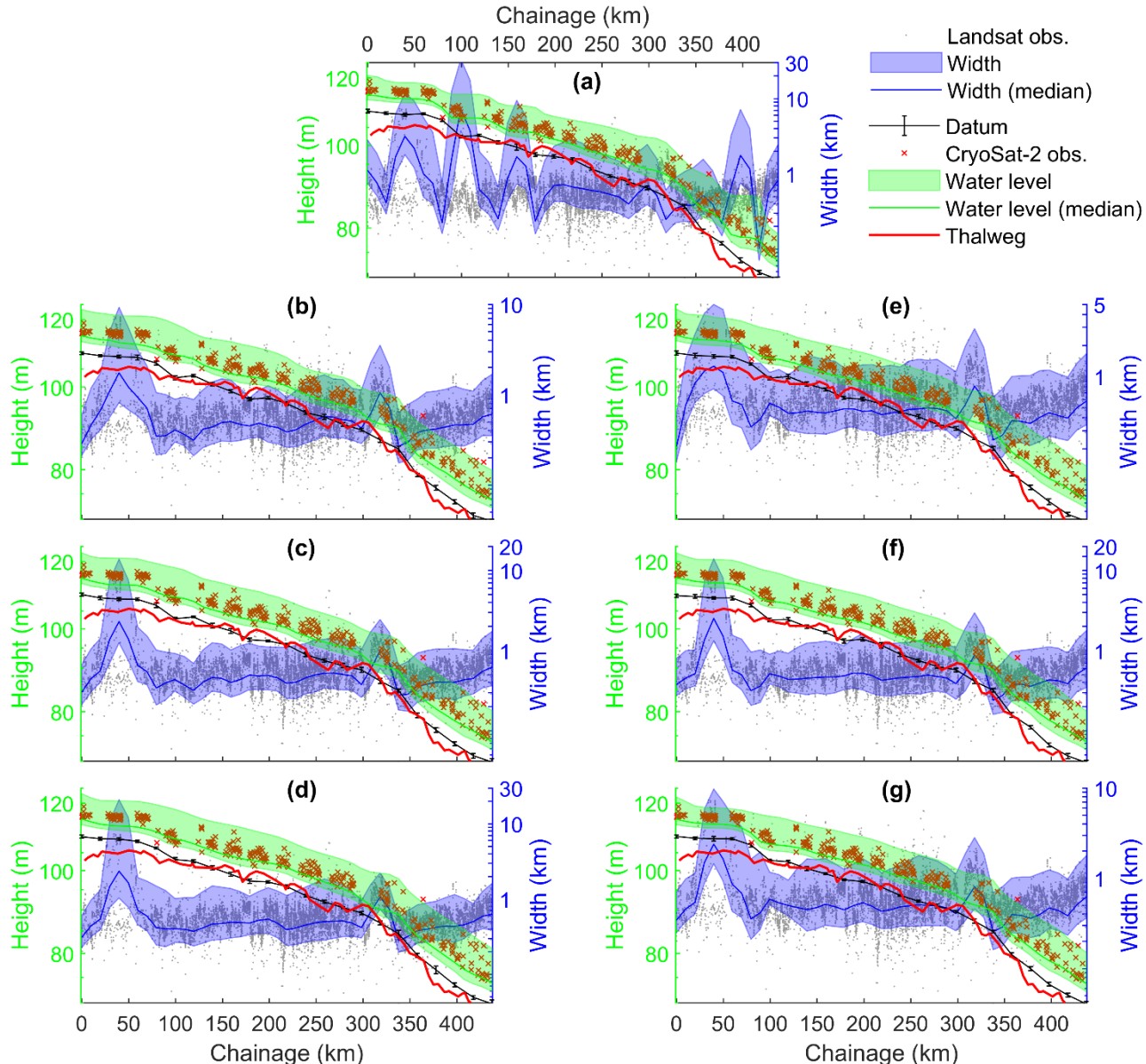

**Figure C2.** Comparison of model simulated water level, width, and datum using different calibration data sets. (a) calibration #1; (b) calibration #2a; (c) calibration #2b; (d) calibration #2; (e) calibration #3a; (f) calibration #3b and (g) calibration #3. Note, y axes are in log scale. Color bands indicate the boundary (i.e. maximum and minimum) of simulations. Along with model simulations, satellite observations of WSE and width are plotted.

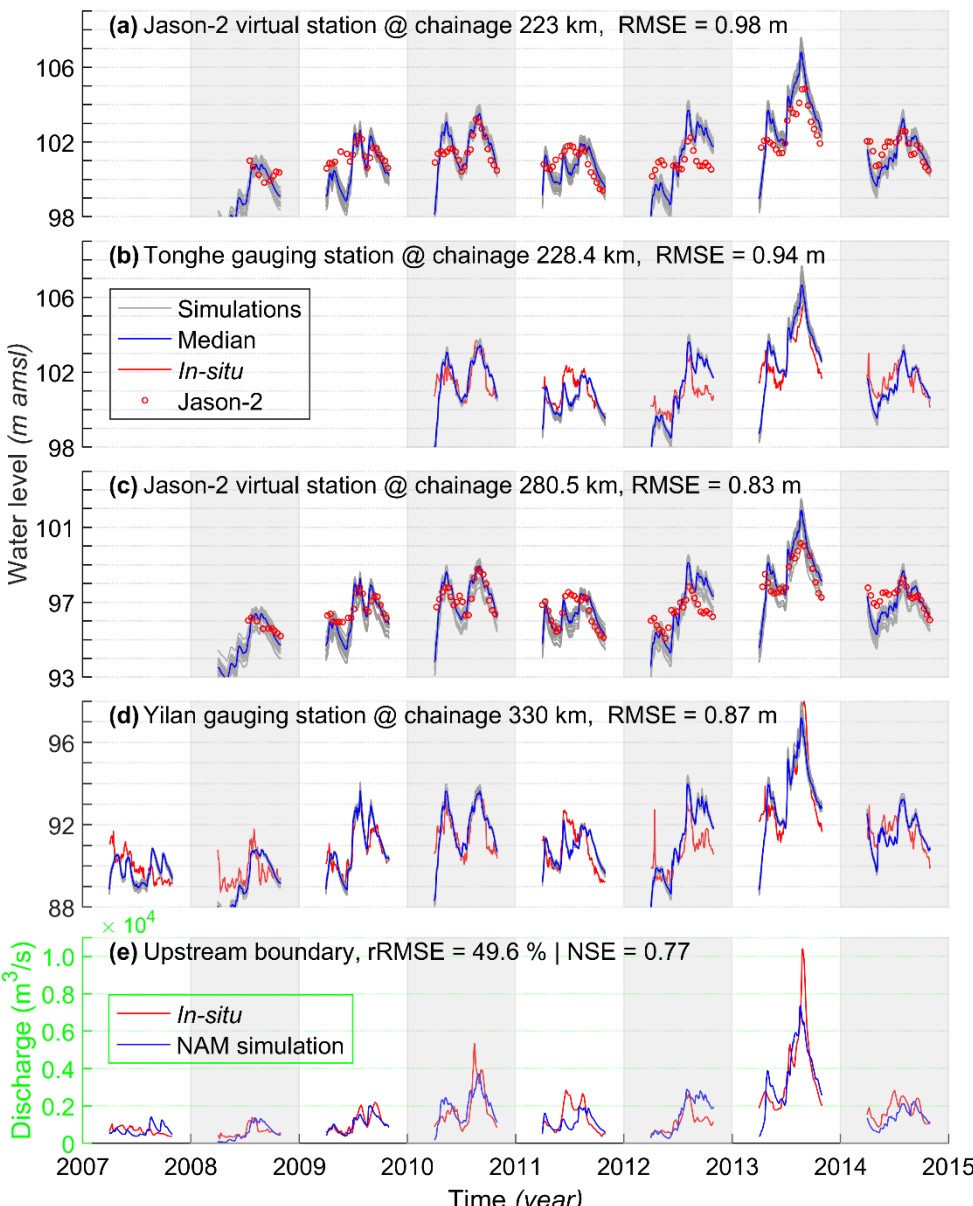

Figure C3. Similar to Figure 6, but upstream boundary is rainfall-runoff model simulation instead of in-situ discharge, which are shown in (e). Note that, NAM model was not calibrated at this location.

*Code and data availability.* River widths were processed on the Google Earth Engine platform. CryoSat-2 were from the study of Jiang et al. (2019). Jason-2 data were downloaded from AVISO+ (https://www.aviso.altimetry.fr/en/data/data-access/ftp.html). Cross-section data of the Changjiang, Songhua, and Yellow rivers were excerpted from the Hydrological Yearbook issued by the Ministry of Water Resources, China. Cross-section data of the Po river are publicly available from the Interregional agency for the river Po (http://geoportale.agenziapo.it/web/index.php/it/?option=com_aipografd3). Data of the Danish rivers (Åmose, Velje) were kindly provided by WSP (http://www.hydrometri.dk/hyd/). The main Matlab scripts are publicly available on Zenodo (https://zenodo.org/record/5555472).

*Author contributions.* SWC developed the methodology in an early stage with inputs from LJ and PBG. LJ further developed the methodology, did the data curation, and performed the calibration work. LJ prepared the manuscript. All authors contributed to editing the manuscript.

*Competing interests.* The authors declare that they have no conflict of interest.

*Acknowledgements.* The authors would like to thank AVISO+, WSP and AIPo for making the data sets available.

*Financial support.* This work is jointly funded by the Danida Fellowship Centre (EOForChina project, File number: 18-M01-DTU) and the Innovation Fund Denmark (ChinaWaterSense project, File number: 8087-00002B).

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
