# Peer review of "Calibrating 1D hydrodynamic river models in the absence of cross-sectional geometry: A new parameterization scheme"

_Hydrology and Earth System Sciences, 2021_

## Referee Comment (RC1)

Dear Editor Albrecht Weerts,

Thank you for sending me "Calibrating 1D hydrodynamic river models in the absence of cross-sectional geometry: A new parameterization scheme" by Jiang et al for review. The paper is overall well written, and the figures and the language of the document are of high quality. When looking at the abstract, I was really enthusiastic, due to my interest for ungauged catchments. However, while reading the paper, my enthusiasm gradually ebbed away, as the approach seems neither novel, nor superior to existing ones. The authors should revise the paper, to convince the reader about the advantages of their approach. The 1D model calibration needs to be described more in detail. I had to guess what the authors actually did. Moving some of the material from the supplement into the manuscript would help the reader.

Kind regards

**Major**

- Novelty
  The proposed method is not novel. Power laws (log linear relations) are just fit to the cross sectional area and conveyance. Furthermore, a fit to the conveyance can be shown to be algebraically identical to fitting the roughness coefficient:

$$A = a\,d^{\beta} \tag{1}$$

$$K = \gamma\,d^{\delta} = \frac{1}{n}A\,R^{2/3} \tag{2}$$

$$\text{with } R \approx d \tag{3}$$

$$K = c\,d^{\delta} = \frac{1}{n}\,A\,d^{2/3} \tag{4}$$

$$= c\,d^{\delta} = \frac{1}{n}\,a\,d^{\beta+2/3} \tag{5}$$

$$\Rightarrow n = \frac{a}{c}\,d^{\beta-\delta+2/3} \tag{6}$$

$$= \tilde{b}\,d^{\tilde{\delta}} \tag{7}$$

Furthermore, Manning's relation for n resembles itself just is a power law to determine the Chezy coefficient C:

$$n = \frac{1}{C}\,d^{-1/6}$$

This is the essence of *Manning et al.* (1890).

- Model and performance

    - The two power laws for A and K have together four coefficients. The authors introduce two more coefficients into A and claim that the extended model superior (line 137). However, for almost any model, introducing more parameters will improve the fit to the calibration data, but it will lead to overparametrization and worse model performance during prediction. The authors should demonstrate that the more complex model performs better. This could be done on hand of the Akaike or Bayesian information criterion, or maybe in a less mathematically robust calibration-validation approach.
    - I do not think that the model performs better, as A and K are still determined by the same log linear relations from d. Thus the model is ill conditioned without need, requiring strong regularization. I can think of other 6 parameter models which could potentially perform better, for example

    $$log(A) = c_0 + c_1 log(d) + c_2 \log(d)^2$$

    - As the first author (Jiang 2019) already published a previous study where the roughness coefficient was fit to the same river reach, it would be interesting to see a direct comparison.
    - Area-Depth relations can also be obtained from satellite images at different stages. How does the method compare in performance?

- Parameter choice
  The authors assume the slope, and thus the bed level, to be known (line 91). From my personal experience, in large ungauged lowland rivers, it is the bed level and thus the slope, which is often the largest factor of uncertainty and much more difficult to determine, even with field measurements. Roughness typically varies little between rivers (*Latrubesse*, 2008), while width can be much sensed remotely much easier than levels. Virtual gauging stations from satellites are typical 100-200 river km apart, and neighbouring virtual gauging stations are not passed simultaneously, introducing uncertainties in the slope estimates due to changes of the hydrograph between passages of the satellite. It would be good to get practical advice on how the slope can be determined for an ungauged river, and how this uncertainty compares to the uncertainty of the A-d and K-d relation.

- Model calibration (line 150ff)
  A lot of information essential to understanding is missing here. The points below are purely guesswork by me and should be explained in the manuscript:

- There is probably a 1D dynamic wave mode for each river, with K and A determined from the flow depth by the power-laws. There are thus two models, the 1D hydrodynamic model (Mike 1D) and a model to predict K and A used by the hydrodynamic model, this should be clarified much earlier in the manuscript, at least when stating the SWE (eq 1,2)
  - Parameters seem to be defined locally for each reach, and to vary along and between rivers.
  - The Levenberg-Marquard is used, but how are the derivatives and the Jacobian calculated? If parameters are defined on reach base, then there will be 100ds of parameters to calibrate, requiring hundreds of hydrodynamic model-runs to alone compute the gradient during one optimization step. There should be a note on the computational effort.
  - What is the value of lambda?
  - $\emptyset$ seems to be the objective function, but it is not defined,

- Results and presentation
  Fits are presented on log-log plots (e.g. Figure 1). This visually emphasizes low-flows, which might be meaningful for drought analysis but it is not suitable for flood risk estimates. Some plots in linear space would thus be insightful.

- Discussion Limitations of the method and sources of error should be discussed, and they should be connection to the physical processes.

  - The model cannot reproduce the hysteresis, i.e. different d-K and d-A relations during the rising and the falling limb of the hydrograph, caused by the dynamic wave. This is in particular the case for low sloping lowland rivers (*Hidayat et al.*, 2011), and strongly sloping mountain reaches.
  - Beforms dynamics can likewise introduce hysteresis and non-uniqueness in the relation between roughness (conveyance) and discharge (depth) (*Cisneros et al.*, 2019).
  - Many large rivers are anastamosing (Irrawaddy, Amazon) and consist of compound channels, and a log-linear relationship will probably not perform well there. The yellow river in the dataset shows this behaviour as well.

**Minor**

73 eq 1,2

- The authors state the shallow water equations (SWE), but the equations used later (3,4,6-10, figure-1) are based on a kinematic wave approach, otherwise the K-d, K-A relation would vary in time. This limitation should be mentioned here and later be addressed in the discussion.

86 "K is much more sensitive to A" → "the model is much more sensitive to K?"
Even with this clarification, the statement seems to be a fallacy, as according to eq 4 and 14, K co-varies with A.

77, 90 "unknowns" is too vague here.

- Variables (A and Q) and parameters (n, $S_0$) should be distinguished
- Furthermore the set of "unknowns" is not minimal, A can be expressed as a function of d, and $S_f$ as a function of Q, given the appropriate relations.

130 "valid at river reach scale instead of individual cross sections"
This is an interesting practical aspect, as the thalweg can vary along a single sharp bend much stronger than the surface elevation varies over hundreds of kilometres. Did the authors sample the area in straight reaches between bends to avoid perturbations due to scours in channel bends, or did they average continuous bathymetry along a river?

S1 This section should be moved the manuscript, to help the reader understand the field site better.

S2 What is $z$? Bed level? So after all, the bed level (slope) is a model parameter which is fit together with the other parameters. This is essential and should be mentioned in the manuscript.

S2 I think this section, or at least parts of it, should also go into the main manuscript, to help the reader understand the model calibration.

**Typos and suggestions**

32 a limited → only a limited

55 there is not just spatial variation, but also temporal variation, c.f. comment on the discussion

85 Sf → $S_f$

149 starting models → starting points?

165  remove somewhat

165  The paper is short, why not moving the map and some other illustrations from the supporting information into the paper?

- Punctuation at the end of equations is missing (, and .).

**References**

Cisneros, J., J. Best, T. van Dijk, and E. Mosselman, Dune morphology and hysteresis in alluvial channels during long-duration floods revealed using high temporal-resolution MBES bathymetry, in *Proceedings International Conference on Marine and River Dune Dynamics–MARID VI*, pp. 1–3, 2019.

Hidayat, H., B. Vermeulen, M. G. Sassi, P. J. J. F. Torfs, and A. J. F. Hoitink, Discharge estimation in a backwater affected meandering river, *Hydrology and Earth System Sciences*, *15*(8), 2717–2728, 2011.

Latrubesse, E. M., Patterns of anabranching channels: The ultimate end-member adjustment of mega rivers, *Geomorphology*, *101*(1-2), 130–145, 2008.

Manning, R., J. P. Griffith, T. Pigot, and L. F. Vernon-Harcourt, *On the flow of water in open channels and pipes*, 1890.

---

## Referee Comment (RC2)

Review Comments

[General Comments]

This manuscript proposes a new concept to calibrate the channel parameters in hydrodynamic models, focusing on flow area and conveyance. The proposed method has a potential to be widely used, as it has less parameters to be calibrated compared to previous approaches, and the calibration works with satellite measurements.

However, I feel the manuscript requires substantial edits before acceptance. Especially, it describes the new methods very well, while its usefulness cannot be assessed because the new method is not compared by previous methods. I suggest to include "comparison to previous method" in the manuscript, to highlight the validity of the proposed method.

Related to above point, the current abstract only describes the idea on using flow-area and conveyance instead of the explicit channel cross-section and roughness parameters, and it does not contain any explanation on the result of experiments to test the applicability of the idea. The abstract should contain the summary of the "case study" part (and also some comparison to previous method). Without the explanation of the case study results, readers cannot guess whether the proposed idea is valid/useful or not. Please consider how to organize the abstract.

Also, I assume this manuscript was once prepared as a letter. I feel the information in the main text was a bit limited, and some mode description can be added (or moved from Supplement), as HESS is the full paper journal. A more detailed explanation in the main text must improve the readability of the manuscript.

[Specific Comments]

P1.L14:

> However, strong correlations appear between cross-section shape parameters and hydraulic roughness in a hydraulic inversion approach.

What authors want to state from this sentence is not very clear. A bit more well-organized explanation is needed. I guess the authors intended to say cross-section and roughness are calibrated independently in previous studies, while they propose a new method which can handle them in a combined manner. I think the explanation can be improved by modifying this sentence together with the previous sentence.

P1.L14:

> That reduces ambiguity

I wonder whether "reducing ambiguity" is a proper word to describe the authors intention or not. I feel the proposed method will "allow some ambiguity" by abiding explicit channel-shape representation, while providing reasonable approximations by using flow area and conveyance (which are more conceptual/ambiguous compared to the cross-section shape and roughness).

P1. L17:

> thus are assumed to be linearly related

The logic is not clear here. Even though the flow area and conveyance can be expressed by power-low functions at each cross-section, there is no reason that they can be linearly related at reach scale. The linear relationship is confirmed by the observations (and not derived from the analysis of the equations), I think "thus are assumed to be" is not logically explained.

P1. L18:

> Data from a wide range of river systems show that the linearity approximation is globally applicable

The relationship is confirmed in 6 rivers, but I cannot get any analysis to support the statement that the similar relationship can be found in other rivers in the globe. I think more data analysis should be added to support this statement.

Alternatively, the authors may discuss why the linear relationship can be found along the river reach. I feel this relationship might have some similarity to AMHG approach (which assume along-reach relationship in cross-section parameters), and they analyzed why and when the relationship can be found by mathematical analysis (Brinkerhoff et al., 2019). I guess this is beyond the scope of this manuscript, but at least the authors should include some discussion on potential reasons on why linear relationship along reach is found.

<ref>

Brinkerhoff, C. B., Gleason, C. J., & Ostendorf, D. W. (2019).

Reconciling at-a-station and at-many-stations hydraulic geometry through river-wide geomorphology. *Geophysical Research Letters*, 46, 9637– 9647. https://doi.org/10.1029/2019GL084529

P2.L57:

> strong parameter correlation appears between cross section shape (wetted perimeter) and hydraulic roughness

It's better to provide some reference related to this statement. I'm not sure whether this is widely known/accepted by hydrology community.

P2.L63

> this ambiguity in hydraulic inverse problems

Please think about another word to replace "ambiguity" as explained above.

P5.L25:

> Due to the linear nature of logarithmic pairs of (A, d) and (K, d),

I cannot fully get the logic behind this statement. Why at-a-station parameters can be correlated as at-many-station parameters (i.e. reach scale relationship)? Is there any mathematical or geographical background reason to support this derivation?

P7.L177:

> the residuals between model predictions and observations (i.e. water level and width);

It is not clear how the width is calculated in the model (as the model now only has flow-area without explicit width representations). How width and flow area can be related? Please explain in a detailed manner.

P7.L163

> Case study

I think it is better to explain the method (experimental setting) of the case study in the method section, and explain in a more detailed manner by moving descriptions from Supplement to main text. As HESS is a full paper journal, it is obviously better to write the minimum explanations on the experiment settings to follow the results.

P8.L182

> Figure 3

Please inform that the "km" indicates the distance from the upstream boundary. It is not clear which is upstream/downstream from the current explanations.

P8.L186:

"Figure 4" should be "Figure 3".

P8.L191:

> the best match with the observed cross sections.

It is not so obvious that the calibration #3 is the best match, as we can observe some exceptions in some reaches. Please provide some objective measure (e.g. RMSE of fitting) to support this statement.

P9.L192:

> compared to less dense one (one per 5 km, Fig. S8) although high-resolution imagery (30 m) can provide plenty of width observations.

This explanation appears suddenly, and it is difficult to follow. I suggest that the authors describe how width observations is prepared in the method section (i.e. how it was sampled), and make an analysis on the sampling sensitivity in the discussion section (not in the case study section) to improve the readability.

P9.L196:

We can see the proposed method worked with a certain accuracy, but it is difficult to guess how much it is useful compared to other/previous methods. Please include some comparison with previous methods (e.g. comparison to the method by Jiang et al. 2019). Can you say the result is improved? Or can you say the result does not differ so much even though the new calibration requires less parameter numbers?

P10.L201:

> the accuracy of simulation is acceptable and comparable to what was achieved using a different approach (Jiang et al., 2019).

Please explain

1) What was the characteristics of the different method? (If not explained, this sentence has no meaningful explanation). I suppose the Jiang et al 2019 calibration was "explicit cross-section and roughness parameters for each section" and thus have "more parameter numbers to be calibrated".

2) Please provide some numbers for subjective comparisons. For example, how the RMSE values are different in these two methods? Without the subjective analysis, I cannot get how the authors concluded that "accuracy is acceptable and comparable".

P11.L249:
> the relationship is generally independent of rivers
I think this is still the observation-based finding, and needs for some theoretical/geographical analysis should be discussed (in the discussion section).

Supplement:
Some contents are better to be moved to main text, such as Test S1, (a part of) Text S3 (except for sampling sensitivity experiments), Figure S5.
* * *
Dai Yamazaki
The University of Tokyo

---

## Author Comment (AC2)

Dear Prof. Yamazaki,

Thank you for reviewing our work and providing useful comments. In the following we will briefly reply (in blue) to your comments and suggestions. A more detailed reply will be provided along with the revision of our manuscript.

[General Comments]

This manuscript proposes a new concept to calibrate the channel parameters in hydrodynamic models, focusing on flow area and conveyance. The proposed method has a potential to be widely used, as it has less parameters to be calibrated compared to previous approaches, and the calibration works with satellite measurements.

However, I feel the manuscript requires substantial edits before acceptance. Especially, it describes the new methods very well, while its usefulness cannot be assessed because the new method is not compared by previous methods. I suggest to include "comparison to previous method" in the manuscript, to highlight the validity of the proposed method.

Related to above point, the current abstract only describes the idea on using flow-area and conveyance instead of the explicit channel cross-section and roughness parameters, and it does not contain any explanation on the result of experiments to test the applicability of the idea. The abstract should contain the summary of the "case study" part (and also some comparison to previous method). Without the explanation of the case study results, readers cannot guess whether the proposed idea is valid/useful or not. Please consider how to organize the abstract.

Also, I assume this manuscript was once prepared as a letter. I feel the information in the main text was a bit limited, and some model description can be added (or moved from Supplement), as HESS is the full paper journal. A more detailed explanation in the main text must improve the readability of the manuscript.

Reply: Thank you for the positive comments. We will add a more detailed comparison of the results to those obtained using previous method, and expand the abstract according to your suggestions.

Agree, we will also expand the main text by moving some details from the supplement.

[Specific Comments]

P1.L14: > However, strong correlations appear between cross-section shape parameters and hydraulic roughness in a hydraulic inversion approach.

What authors want to state from this sentence is not very clear. A bit more well-organized explanation is needed. I guess the authors intended to say cross-section and roughness are calibrated independently in previous studies, while they propose a new method which can handle them in a combined manner. I think the explanation can be improved by modifying this sentence together with the previous sentence.

Reply: Simultaneous parameterization of channel geometry and roughness is very difficult, and often there are some trade-offs between them. Instead of calibrating two sets of parameters, we propose a new method that combine them into one parameter, i.e. conveyance. Yes, we will revise this sentence to make it clear.

P1.L14: > That reduces ambiguity

I wonder whether "reducing ambiguity" is a proper word to describe the authors intention or not. I feel the proposed method will "allow some ambiguity" by abiding explicit channel-shape representation, while

providing reasonable approximations by using flow area and conveyance (which are more conceptual/ambiguous compared to the cross section shape and roughness).

Reply: Our new approach does not need to consider the cross-sectional shape and roughness anymore. In this sense, we do not have ambiguity issue. To clear up this doubt, we will rephrase this sentence.

P1. L17:

> thus are assumed to be linearly related

The logic is not clear here. Even though the flow area and conveyance can be expressed by power-low functions at each cross-section, there is no reason that they can be linearly related at reach scale. The linear relationship is confirmed by the observations (and not derived from the analysis of the equations), I think "thus are assumed to be" is not logically explained.

Reply: Yes, agree with you. We will revise these sentences accordingly.

P1. L18:

> Data from a wide range of river systems show that the linearity approximation is globally applicable

The relationship is confirmed in 6 rivers, but I cannot get any analysis to support the statement that the similar relationship can be found in other rivers in the globe. I think more data analysis should be added to support this statement.

Alternatively, the authors may discuss why the linear relationship can be found along the river reach. I feel this relationship might have some similarity to AMHG approach (which assume along-reach relationship in cross-section parameters), and they analyzed why and when the relationship can be found by mathematical analysis (Brinkerhoff et al., 2019). I guess this is beyond the scope of this manuscript, but at least the authors should include some discussion on potential reasons on why linear relationship along reach is found.

Reply: Agree. Considering the difficulties to collect surveyed cross-sectional data sets, we will add some discussions and call for validations.

P2.L57:

> strong parameter correlation appears between cross section shape (wetted perimeter) and hydraulic roughness

It's better to provide some reference related to this statement. I'm not sure whether this is widely known/accepted by hydrology community.

Reply: We had calculated the correlation coefficients of roughness and river width and showed strong correlations in our previous study. We will add the reference.

P2.L63

> this ambiguity in hydraulic inverse problems

Please think about another word to replace "ambiguity" as explained above.

Reply: Will rephrase this sentence.

P5.L25:

> Due to the linear nature of logarithmic pairs of (A, d) and (K, d),

I cannot fully get the logic behind this statement. Why at-a-station parameters can be correlated as at-many-station parameters (i.e. reach scale relationship)? Is there any mathematical or geographical background reason to support this derivation?

Reply: Take the linear relationships of (A, d) and (K, d) as the start,

$$log\ A(x,t) = \alpha(x) + \beta(x)\ log\ d(x,t) \tag{9}$$
$$log\ K(x,t) = \gamma(x) + \delta(x)\ log\ d(x,t) \tag{10}$$

Substituting log $d$ in Eq (9), one can get

$$log\ A = \alpha + \frac{\beta}{\delta}(\log K - \gamma) \tag{R1}$$

By rearranging Eq (R1), we have

$$\alpha = \log A - \frac{\beta}{\delta}\log K + \frac{\beta}{\delta}\gamma \tag{R2}$$

Similarly, we divide Eq (6) by Eq (7), we can obtain

$$\frac{A}{K} = \frac{a}{c}d^{\beta-\delta} \tag{R3}$$

Taking log transformation and rearranging the equation, that leaves

$$\beta = \frac{\log A - \log K + \gamma - \alpha}{\log d} + \delta \tag{R4}$$

The above derivations (Eqs R2 and R4) are for each individual cross sections. However, because on the right side both equations have A and K that are dependent on water level, the intercept of linear equations (R2) and (R4) are not just constant.

One should not confuse Eq (R2) and Eq (R4) with Eq (11) and Eq (12) that are

$$\alpha = p_1 + p_2\gamma \tag{11}$$

$$\beta = p_3 + p_4\delta \tag{12}$$

These two equations are derived by fitting a linear function to observed data at the reach scale as shown in Figure 2. Because we see it in our dataset, we then use it to simplify the hydraulic inverse problem by tying some of the parameters together, i.e. reducing the number of fitting parameters. Due to the spatial variation of α, γ, β and δ, we have to constrain 4*N of parameters (N is the number of cross sections). By using equations 11 and 12, we can halve the number of parameters.

We will expand this section to clear up the doubts.

P7.L177:

> the residuals between model predictions and observations (i.e. water level and width);

It is not clear how the width is calculated in the model (as the model now only has flow-area without explicit width representations). How width and flow area can be related? Please explain in a detailed manner.

Reply: In the model, the width is calculated as the derivative of flow area and depth. Will add this in the model setup section.

P7.L163

> Case study

I think it is better to explain the method (experimental setting) of the case study in the method section, and explain in a more detailed manner by moving descriptions from Supplement to main text. As HESS is a full paper journal, it is obviously better to write the minimum explanations on the experiment settings to follow the results.

Reply: Will expand this section by moving materials from the supplement.

P8.L182

> Figure 3

Please inform that the "km" indicates the distance from the upstream boundary. It is not clear which is

upstream/downstream from the current explanations.

Reply: Thank you for pointing this out. Will explain it in the caption.

P8.L186:

"Figure 4" should be "Figure 3".

Reply: Yes, will revise it.

P8.L191:

> the best match with the observed cross sections.

It is not so obvious that the calibration #3 is the best match, as we can observe some exceptions in some reaches. Please provide some objective measure (e.g. RMSE of fitting) to support this statement.

Reply: Thank you. We will calculate the statistics.

P9.L192:

> compared to less dense one (one per 5 km, Fig. S8) although high-resolution imagery (30 m) can provide plenty of width observations.

This explanation appears suddenly, and it is difficult to follow. I suggest that the authors describe how width observations is prepared in the method section (i.e. how it was sampled), and make an analysis on the sampling sensitivity in the discussion section (not in the case study section) to improve the readability.

Reply: Agree, we had a text explaining the procedure for preparing width observations in the supplement. We will move the related materials to main text.

P9.L196:

We can see the proposed method worked with a certain accuracy, but it is difficult to guess how much it is useful compared to other/previous methods. Please include some comparison with previous methods (e.g. comparison to the method by Jiang et al. 2019). Can you say the result is improved? Or can you say the result does not differ so much even though the new calibration requires less parameter numbers?

Reply: Yes, the results are very similar, but this method does not need to assume any specific shape of the cross sections. We will further explain this and include more details.

P10.L201:

> the accuracy of simulation is acceptable and comparable to what was achieved using a different approach (Jiang et al., 2019).

Please explain

1) What was the characteristics of the different method? (If not explained, this sentence has no meaningful

explanation). I suppose the Jiang et al 2019 calibration was "explicit cross-section and roughness parameters for each section" and thus have "more parameter numbers to be calibrated".

Reply: The major difference is whether considers the cross-sectional shape or not. Our previous study assumed a certain shape (rectangle) while this one does not need to consider shape anymore.

We will make it clearer in the text.

2) Please provide some numbers for subjective comparisons. For example, how the RMSE values are different in these two methods? Without the subjective analysis, I cannot get how the authors concluded that "accuracy is acceptable and comparable".

Reply: Will added the RMSE values of both methods.

P11.L249:

> the relationship is generally independent of rivers

I think this is still the observation-based finding, and needs for some theoretical/geographical analysis should be discussed (in the discussion section).

Reply: We will discuss this point further.

Supplement:

Some contents are better to be moved to main text, such as Test S1, (a part of) Text S3 (except for sampling sensitivity experiments), Figure S5.

Reply: We will adopt your suggestions.

---

## Author Response (AR1)

**Replies to the comments by anonymous referee #1**

We thank the reviewer for spending time and effort reviewing our work. Below the reviewer's comments are in black and the replies in blue.

Major

- Novelty

The proposed method is not novel. Power laws (log linear relations) are just fit to the cross sectional area and conveyance. Furthermore, a fit to the conveyance can be shown to be algebraically identical to fitting the roughness coefficient:

$$A = ad^\beta \tag{1}$$

$$K = cd^\delta \tag{2}$$

$$\text{with } R \approx d \tag{3}$$

$$K = cd^\delta = \frac{1}{n}Ad^{2/3} \tag{4}$$

$$= cd^\delta = \frac{1}{n}ad^{\beta+2/3} \tag{5}$$

$$\Rightarrow n = \frac{a}{c}d^{\beta-\delta+2/3} \tag{6}$$

$$= \tilde{b}d^{\tilde{\delta}} \tag{7}$$

Furthermore, Manning's relation for n resembles itself just is a power law to determine the Chezy coefficient C:

$$n = \frac{1}{C}d^{-1/6}$$

This is the essence of *Manning et al. (1890).*

Reply: we partially agree with the statement.

Firstly, we should point out that $R \approx d$ is an approximation that is valid for wide rectangular channels and does not necessarily hold for more complex cross-sectional shapes. So, in the general case, conveyance integrates both Manning roughness and cross-sectional shape parameters.

We agree that the power-law relationships between flow area / conveyance and flow depth are not new. The novelty is to use these relationships in a hydraulic inversion, i.e. estimating the parameters of these relationships by fitting simulated water surface elevations/ river widths to observed water surface elevations/ river widths from satellite data.

Previously, most studies focusing on hydraulic parameter estimation using satellite water surface elevation data, if not all, have used surveyed bathymetry data or parameterized the geometry of channels to calculate flow area and conveyance. This new parameterization bypasses this difficulty and thus does not require the assumption of any specific crosssectional shape. Therefore, this approach is new and fundamentally different from previous studies. This has been clearly described in the Introduction section.

- Model and performance

→ The two power laws for A and K have together four coefficients. The authors introduce two more coefficients into A and claim that the extended model superior (line 137). However, for almost any model, introducing more parameters will improve the fit to the calibration data, but it will lead to overparameterization and worse model performance during prediction. The authors should demonstrate that the more complex model performs better. This could be done on hand of the Akaike or Bayesian information criterion, or maybe in a less mathematically robust calibration-validation approach.

Reply: The reason to introduce four coefficients, i.e. p1 ~ p4, is to reduce the number of parameters to be calibrated. Originally, as shown in Eq. (9) and Eq. (10), there are four parameters, i.e. α, β, γ, δ, which are varying with chainage (i.e. river coordinate). That is, there are totally 4*XS parameters (XS is the number of cross sections). Because of the correlation between α and γ (also β and δ) at reach scale, four spatially constant coefficients can be introduced. Therefore, the number of parameters is reduced to (2*XS+4) as shown in Eq. (13) and Eq. (14).

→ I do not think that the model performs better, as A and K are still determined by the same log linear relations from d. Thus the model is ill conditioned without need, requiring strong regularization. I can think of other 6 parameter models which could potentially perform better, for example

$$log(A) = c_0 + c_1 log(d) + c_2 log(d)^2$$

Reply: If we use 6 parameter models for each cross section, we will have (6*XS) parameters to calibrate, which is probably very unrealistic.

As explained above, the introduction of four coefficients is simply to reduce the number of free calibration parameters. This improves the stability of the hydraulic inverse problem.

→ As the first author (Jiang 2019) already published a previous study where the roughness coefficient was fit to the same river reach, it would be interesting to see a direct comparison.

Reply: Yes, we have performed a directly comparison of model simulations. In the revised manuscript, we plotted water level simulations from both methods, and the RMSEs are reported. Below is the modified figure.

[Figure]

Figure R1. The same as Figure 6 in the main text. Validation of water level at four stations. Simulations (grey lines) and median (dash-line) are results from the proposed method in this study, and simulations (S1) (red line) are from previous study using standard calibration, i.e. simultaneously calibrating both cross-sectional shape parameters and roughness.

$\rightarrow$ Area-Depth relations can also be obtained from satellite images at different stages. How does the method compare in performance?

Reply: Please note that we used both river width and water surface elevation data in the inverse problem. We have conducted an experiment (Table below) to investigate the power of different data sets in constraining model parameters. Two methods (river width only and WSE only) show similar performance in terms of RMSE of WSE at two gauging stations as shown in below. However, if a channel is embanked, for instance, model parameters may not be sensitive to the small changes of river width. Comparatively, WSE would be more informative in this case. This has been discussed at the beginning of the Discussion section.

Table R1. The same as Table 1 in the main text, describing the calibration scenarios.

| Scenario | Description | Num. of WSE | Num. of width |
|---|---|---|---|
| Calibration #1 | Calibration with WSE observations only | 261 | 0 |
| Calibration #2a | Calibration with one width per 5 km | 0 | 88 |
| Calibration #2b | Calibration with one width per 2 km | 0 | 219 |
| Calibration #2 | Calibration with width observations only | 0 | 10022 |
| Calibration #3a | Calibration with WSE and one width per 5 km | 261 | 88 |
| Calibration #3b | Calibration with WSE and one width per 2 km | 261 | 219 |
| Calibration #3 | Calibration with WSE and width observations | 261 | 10022 |

[Figure]

Figure R2. The same as Figure 5 in the main text. Boxplots of evaluations of simulated water level against in-situ gauging records at two gauging stations. Calibration scenarios (refer to Table R1) indicated on the x-axis.

- Parameter choice

The authors assume the slope, and thus the bed level, to be known (line 91). From my personal experience, in large ungauged lowland rivers, it is the bed level and thus the slope, which is often the largest factor of uncertainty and much more difficult to determine, even with field measurements. Roughness typically varies little between rivers (Latrubesse, 2008), while width can be much sensed remotely much easier than levels. Virtual gauging stations from satellites are typical 100-200 river km apart, and neighbouring virtual gauging stations are not passed simultaneously, introducing uncertainties in the slope estimates due to changes of the

hydrograph between passages of the satellite. It would be good to get practical advice on how the slope can be determined for an ungauged river, and how this uncertainty compares to the uncertainty of the A-d and K-d relation.

Reply: The slope is not assumed to be known. For each cross section, there is a parameter Z0, representing the local datum, i.e. river bed elevation. Bed slope is then determined from the difference of adjacent bottom elevations.

Regarding the distance of virtual stations, yes, it is about 80 km for Envisat while 315 km for Jason series. But, it is about 7 km for CryoSat-2 data. Therefore, we can obtain more information from densely distributed CryoSat-2 derived WSE data. The advantages of this kind of altimetry datasets have been reported by several studies (Domeneghetti et al., 2021; Jiang et al., 2019; Schneider et al., 2018).

- Model calibration (line 150ff)

   A lot of information essential to understanding is missing here. The points below are purely guesswork by me and should be explained in the manuscript:

→  There is probably a 1D dynamic wave mode for each river, with K and A determined from the flow depth by the power-laws. There are thus two models, the 1D hydrodynamic model (Mike 1D) and a model to predict K and A used by the hydrodynamic model, this should be clarified much earlier in the manuscript, at least when stating the SWE (eq 1,2)

Reply: During the calibration, the second model you refer to is an integral part of the Mike 1D hydraulic model. That is, the new relationships d-A and d-K, which change in each iteration of the inversion, are updated in the Mike 1D model. Therefore, we have only one model. We have added a few sentences to avoid this confusion.

   *"The calibration is implemented in Matlab. C# scripts are used to modify and dump MIKE Hydro River parameters and simulation results. The power-law relationships are an integral part of the MIKE Hydro River model. Specifically, for each iteration of the optimization, the updated parameters by LM algorithm as well as the calculated flow area and conveyance relationships are passed to a C# script that updates the setup of the MIKE Hydro River model. Then the model is executed, and the results are passed on to Matlab."*

→  Parameters seem to be defined locally for each reach, and to vary along and between rivers.

Reply: Correct, $Z_0$, $\gamma$ and $\delta$ are spatially varying, but $p1$, $p_2$, $p_3$, and $p_4$ are constant. This is explicitly described in line 151.

→  The Levenberg-Marquard is used, but how are the derivatives and the Jacobian calculated? If parameters are defined on reach base, then there will be 100ds of parameters to calibrate, requiring hundreds of hydrodynamic model-runs to alone compute the gradient during one optimization step. There should be a note on the computational effort.

Reply: Normally, in each iterative step, the Jacobian of the objective function is calculated or approximated using finite difference. However, computing the Jacobian in every iteration is computational expensive and in some cases the Jacobian does not change and thus, evaluation of the Jacobian can be unnecessarily costly. Instead, Broyden's rank-one update of the Jacobian (Broyden, 1965) is more efficient (Madsen et al., 2004). We use the Immoptibox toolbox (Nielsen and Völcker, 2010) to optimize the objective function.

The number of model runs is around 200. The time consumed for this optimization is about a few hours (1 ~ 4 hours). The calibrations were conducted on a windows server 2016 (Inter® Xeon® Gold 6154 CPU @ 3 GHz, 2993 Mhz) using 4 cores.

$\rightarrow$ What is the value of lambda?

Reply: The regularization parameters, i.e. $\lambda_\gamma$, $\lambda_\delta$, $\lambda_{p1}$, $\lambda_{p2}$, $\lambda_{p3}$, $\lambda_{p4}$ are empirically set as 0.1, 0.1, 0.15, 0.15, 0.15, 0.15, respectively, by trial and error.

A detailed description is given as Text S2 in the supporting information.

$\rightarrow$ $\Phi$ seems to be the objective function, but it is not defined.

Reply: Correct, that is Eq. (15). We will define it in the revision.

- Results and presentation

  Fits are presented on log-log plots (e.g. Figure 1). This visually emphasizes low-flows, which might be meaningful for drought analysis but it is not suitable for food risk estimates. Some plots in linear space would thus be insightful.

  Reply: Yes, but the purpose of this figure is to demonstrate the linear relationship. We will also add a linear scale plot to the supporting information.

- Discussion Limitations of the method and sources of error should be discussed, and they should be connection to the physical processes.

  $\rightarrow$ The model cannot reproduce the hysteresis, i.e. different d-K and d-A relations during the rising and the falling limb of the hydrograph, caused by the dynamic wave. This is in particular the case for low sloping lowland rivers (Hidayat et al., 2011), and strongly sloping mountain reaches.

  Reply: We are not clear about what type of hysteresis the reviewer refers to. Hysteresis in rating curves does not occur because of d-K and d-A relationships but because of the pressure terms in the Saint-Venant equations, which contribute differently in the ascending and descending branches of flood waves. The Mike 1D hydraulic model uses dynamic wave model, thus, is able to simulate hysteresis. It should be noted that we did not directly calculate flow by multiplying conveyance (K) with bed slope ($S_0$).

  $\rightarrow$ Bedforms dynamics can likewise introduce hysteresis and non-uniqueness in the relation between roughness (conveyance) and discharge (depth) (Cisneros et al., 2019).

Reply: Yes, bedform changes will lead to time variable d-A and d-K curves, which our approach does not allow. However, there are many situations where stable bed is a good approximation. We have acknowledged this point in the revision.

> *"Moreover, this approach assumes that the established curves are time invariable, which is not applicable to rivers with significant bedform changes."*

$\rightarrow$ Many large rivers are anastamosing (Irrawaddy, Amazon) and consist of compound channels, and a log-linear relationship will probably not perform well there. The yellow river in the dataset shows this behaviour as well.

Reply: Yes, one linear regression may be not able to describe both the low flow and high flow for certain rivers, especially those with a significant floodplain. We have reported this issue at section 2.2 (lines 110-115) when introducing this approach. Moreover, we further discussed this point in the discussion section.

> *"As we mentioned, this study only focuses on the main channel and does not account for overbank flow. In the presence of significant floodplains, the linearity of the curve may fail at bankfull depth as seen in* **Error! Reference source not found.** *and* **Error! Reference source not found.***.* *Consequently, as seen in* **Error! Reference source not found.***,* *the model over-estimates extreme flood peak (year 2013). Similarly, one curve may not be able to describe anastomosing rivers that consist of compound channels. To solve this problem, a second curve is needed to describe the overbank flow as suggested by Garbrecht (1990)…."*

Minor

The authors state the shallow water equations (SWE), but the equations used later (3,4,6-10, figure-1) are based on a kinematic wave approach, otherwise the K-d, K-A relation would vary in time. This limitation should be mentioned here and later be addressed in the discussion.

Reply: We believe that the reviewer refers to Q-d and Q-A relationships. Those are indeed time variable, both in reality and also in our model, because the model uses the full dynamic wave format of the De Saint Venant equations. However, K-d and K-A relationships are invariable in time, because we assume a static bed geometry and time-invariable Manning roughness. Please note again that we do not assume Q=K (i.e. discharge equal conveyance) anywhere in our approach.

86 "K is much more sensitive to A" ! "the model is much more sensitive to K?" Even with this clarification, the statement seems to be a fallacy, as according to eq 4 and 14, K co-varies with A.

Reply: K is a function of both roughness and area. Even when the flow area is constant, the conveyance can be different depending on the roughness.

The statement is valid for kinematic wave model and diffusive wave model specially. The momentum equation is simply as below when it is in steady state:

$$gA(d)\frac{\partial d}{\partial x} - \underbrace{\underbrace{gA(d)\left(S_0 - \frac{Q^2}{K^2(d)}\right)}_{Kinematic\ wave}}_{Diffusive\ wave} = 0$$

Therefore, A(d) can be eliminated from the equation. In this way, it will not affect the momentum equation.

77, 90 "unknowns" is too vague here.

$\rightarrow$   Variables (A and Q) and parameters (n, S0) should be distinguished.

$\rightarrow$   Furthermore, the set of "unknowns" is not minimal, A can be expressed as a function of d, and Sf as a function of Q, given the appropriate relations.

Reply: We treat Q and d as variable as we try to solve these two quantities, while others are parameters. Therefore, we have revised it accordingly as:

*"Equations (1) and (5) provide two equations with still five unknowns, i.e. two variables (Q and d), and three parameters (A, So, and K)."*

130 "valid at river reach scale instead of individual cross sections"

This is an interesting practical aspect, as the thalweg can vary along a single sharp bend much stronger than the surface elevation varies over hundreds of kilometres. Did the authors sample the area in straight reaches between bends to avoid perturbations due to scours in channel bends, or did they average continuous bathymetry along a river?

Reply: We did not intentionally avoid any bends. The cross sections are evenly distributed at an interval of ~20 km along the centerline of main channel. A detailed map is added in the revised manuscript as below.

[Figure]

Figure R3. The same as Figure 3 in the main text. Study area and the river network.

S1 This section should be moved the manuscript, to help the reader understand the field site better.

Reply: We have integrated supplementary information into the main text.

S2 What is z? Bed level? So after all, the bed level (slope) is a model parameter which is fit together with the other parameters. This is essential and should be mentioned in the manuscript.

Reply: Correct, as explained in previous reply, we did not assume the slope but treat it as a parameter.

S2 I think this section, or at least parts of it, should also go into the main manuscript, to help the reader understand the model calibration.

Reply: We have merged this part into the main text.

Typos and suggestions

32 a limited → only a limited

Reply: We will revise it in the revision.

55 there is not just spatial variation, but also temporal variation, c.f. comment on the discussion

Reply: We will mention this point in the discussion.

85 Sf → $S_f$

Reply: We will revise it accordingly.

149 starting models → starting points?

Reply: Yes, we will revise it.

165 remove somewhat

Reply: Agree.

165 The paper is short, why not moving the map and some other illustrations from the supporting information into the paper? Punctuation at the end of equations is missing (, and .).

Reply: Thank you. We will revise it accordingly.

References

Broyden, C. G.: A class of methods for solving nonlinear simultaneous equations, Math. Comput., 19(92), 577–577, doi:10.1090/S0025-5718-1965-0198670-6, 1965.

Domeneghetti, A., Molari, G., Tourian, M. J., Tarpanelli, A., Behnia, S., Moramarco, T., Sneeuw, N. and Brath, A.: Testing the use of single- and multi-mission satellite altimetry for the calibration of hydraulic models, Adv. Water Resour., 151, 103887, doi:10.1016/j.advwatres.2021.103887, 2021.

Jiang, L., Madsen, H. and Bauer-Gottwein, P.: Simultaneous calibration of multiple hydrodynamic model parameters using satellite altimetry observations of water surface elevation in the Songhua River, Remote Sens. Environ., 225, 229–247, doi:10.1016/j.rse.2019.03.014, 2019.

Madsen, K., Nielsen, H. B. and Tingleff, O.: Methods for Non-Linear Least Squares Problems (2nd ed.)., 2004.

Nielsen, H. B. and Völcker, C.: IMMOPTIBOX: A Matlab Toolbox for Optimization and Data Fitting, 2010.

Schneider, R., Tarpanelli, A., Nielsen, K., Madsen, H. and Bauer-Gottwein, P.: Evaluation of multi-mode CryoSat-2 altimetry data over the Po River against in situ data and a hydrodynamic model, Adv. Water Resour., 112(August 2017), 17–26, doi:10.1016/j.advwatres.2017.11.027, 2018.

**Replies to the comments by referee #2**

Dear Prof. Yamazaki,

Thank you for reviewing our work and providing useful comments. In the following we will briefly reply (in blue) to your comments and suggestions. A more detailed reply is provided after your original comments.

[General Comments]

This manuscript proposes a new concept to calibrate the channel parameters in hydrodynamic models, focusing on flow area and conveyance. The proposed method has a potential to be widely used, as it has less parameters to be calibrated compared to previous approaches, and the calibration works with satellite measurements.

However, I feel the manuscript requires substantial edits before acceptance. Especially, it describes the new methods very well, while its usefulness cannot be assessed because the new method is not compared by previous methods. I suggest to include "comparison to previous method" in the manuscript, to highlight the validity of the proposed method.

Related to above point, the current abstract only describes the idea on using flow-area and conveyance instead of the explicit channel cross-section and roughness parameters, and it does not contain any explanation on the result of experiments to test the applicability of the idea. The abstract should contain the summary of the "case study" part (and also some comparison to previous method). Without the explanation of the case study results, readers cannot guess whether the proposed idea is valid/useful or not. Please consider how to organize the abstract.

Also, I assume this manuscript was once prepared as a letter. I feel the information in the main text was a bit limited, and some model description can be added (or moved from Supplement), as HESS is the full paper journal. A more detailed explanation in the main text must improve the readability of the manuscript.

Reply: Thank you for the positive comments.

- We have added simulations of water level from previous study on top of simulations from this study as shown below. Texts are also revised accordingly.

[Figure]

Figure R1. Same as Figure 6 in the manuscript. Validation of simulated water level (non-frozen periods) at four stations. Simulations (grey lines) and median (dash-line) are results from the proposed method in this study, and simulations (S1) (red line) are from previous study using standard calibration, i.e. simultaneously calibrating both cross-sectional shape parameters and roughness.

- We have added the main results of the case study in the end of the abstract as below:

  *"Its feasibility and performance are illustrated using satellite observations of river width and water surface elevation in the Songhua River, China. Results show that this approach is able to reproduce water level dynamics with root mean square error value of 0.44 m and 0.50 m at two gauging stations, which is comparable to that achieved using a standard calibration approach. In summary, this study puts forward an alternative method to parameterize and calibrate river models using satellite observations of river width and water surface elevation."*

[Specific Comments]

P1.L14: > However, strong correlations appear between cross-section shape parameters and hydraulic roughness in a hydraulic inversion approach.

What authors want to state from this sentence is not very clear. A bit more well-organized explanation is needed. I guess the authors intended to say cross-section and roughness are calibrated independently in previous studies, while they propose a new method which can handle them in a combined manner. I think the explanation can be improved by modifying this sentence together with the previous sentence.

Reply: Simultaneous parameterization of channel geometry and roughness is very difficult, and often there are some trade-offs between them. Instead of calibrating two sets of parameters, we propose a new method that combine them into one parameter, i.e. conveyance. We rephrase it as:

> *"Therefore, river cross-section geometry has commonly been approximated using highly simplified generic shapes. Simultaneous calibration of shape parameters and roughness is difficult, because often there are trade-offs between them. Instead of parameterizing cross-section geometry and hydraulic roughness separately, this study introduces a parameterization of 1D hydrodynamic models by combining cross-section geometry and roughness into one conveyance parameter."*

P1.L14: > That reduces ambiguity

I wonder whether "reducing ambiguity" is a proper word to describe the authors intention or not. I feel the proposed method will "allow some ambiguity" by abiding explicit channel-shape representation, while providing reasonable approximations by using flow area and conveyance (which are more conceptual/ambiguous compared to the cross section shape and roughness).

Reply: Our new approach does not need to consider the cross-sectional shape and roughness separately anymore. In this sense, we do not have any ambiguity issue. To clear up this doubt, we have removed this phrase.

P1. L17:

> thus are assumed to be linearly related

The logic is not clear here. Even though the flow area and conveyance can be expressed by power-low functions at each cross-section, there is no reason that they can be linearly related at reach scale. The linear relationship is confirmed by the observations (and not derived from the analysis of the equations), I think "thus are assumed to be" is not logically explained.

Reply: Yes, we agree with you. Actually, this relationship can be mathematically derived (see the derivations under your comment below) as well as the evidence from observations. We have revised this sentence as below:

> *"Flow area and conveyance are expressed as power-law functions of flow depth, and they are found to be linearly related in log-log space at reach scale."*

P1. L18:

> Data from a wide range of river systems show that the linearity approximation is globally applicable

The relationship is confirmed in 6 rivers, but I cannot get any analysis to support the statement that the similar relationship can be found in other rivers in the globe. I think more data analysis should be added to support this statement.

Alternatively, the authors may discuss why the linear relationship can be found along the river reach. I feel this relationship might have some similarity to AMHG approach (which assume along-reach relationship in cross-section parameters), and they analyzed why and when the relationship can be found by mathematical analysis (Brinkerhoff et al., 2019). I guess this is beyond the scope of this manuscript, but at least the authors should include some discussion on potential reasons on why linear relationship along reach is found.

Reply: Agree. Considering the difficulties to collect surveyed cross-sectional data sets, we have added some discussions and call for validations in the discussion section.

> *"The power-law function relationships of flow area / conveyance and flow depth and the corresponding linearity approximation are confirmed in 6 rivers. One may argue that the relationship may not be globally applicable due to the limited number of rivers to validate the relationships. We cannot rebut this argument without collecting a large sample of surveyed cross sections, which is difficult because of data access problems. However, the rivers we used are of diverse sizes (width ranging from a few meters to kilometers) and flow characteristics, and are from different climate zones (Arctic, Mediterranean, and Asian temperate climates). Therefore, we believe that the relationship holds globally, and we call for extensive validation using other rivers…"*

P2.L57:

> strong parameter correlation appears between cross section shape (wetted perimeter) and hydraulic roughness

It's better to provide some reference related to this statement. I'm not sure whether this is widely known/accepted by hydrology community.

Reply: We had calculated the correlation coefficients of roughness and river width and showed strong correlations in our previous study. We have added the reference, i.e. Jiang et al. (2019).

P2.L63

> this ambiguity in hydraulic inverse problems

Please think about another word to replace "ambiguity" as explained above.

Reply: The hydraulic inverse problem is ill-posed, because changes in Manning roughness can be compensated by changes in bed geometry, which leads to significant parameter correlation. We replaced ambiguity with "parameter correlation".

P5.L25:

> Due to the linear nature of logarithmic pairs of (A, d) and (K, d),

I cannot fully get the logic behind this statement. Why at-a-station parameters can be correlated as at-many-station parameters (i.e. reach scale relationship)? Is there any mathematical or geographical background reason to support this derivation?

Reply: Take the linear relationships of (A, d) and (K, d) as the start,

$$log\ A(x,t) = \alpha(x) + \beta(x)\ log\ d(x,t) \tag{9}$$
$$log\ K(x,t) = \gamma(x) + \delta(x)\ log\ d(x,t) \tag{10}$$

Substituting log *d* in Eq (9), one can get

$$log\ A = \alpha + \frac{\beta}{\delta}(log\ K - \gamma) \tag{R1}$$

By rearranging Eq (R1), we have

$$\alpha = log\ A - \frac{\beta}{\delta}log\ K + \frac{\beta}{\delta}\gamma \tag{R2}$$

Similarly, we divide Eq (6) by Eq (7), we can obtain

$$\frac{A}{K} = \frac{a}{c}d^{\beta-\delta} \tag{R3}$$

Taking log transformation and rearranging the equation, that leaves

$$\beta = \frac{log\ A - log\ K + \gamma - \alpha}{log\ d} + \delta \tag{R4}$$

The above derivations (Eqs R2 and R4) are for each individual cross sections. However, because on the right side both equations have A and K that are dependent on water level, the intercept of linear equations (R2) and (R4) are not just constant.

One should not confuse Eq (R2) and Eq (R4) with Eq (11) and Eq (12) that are

$$\alpha = p_1 + p_2\gamma \tag{11}$$

$$\beta = p_3 + p_4\delta \tag{12}$$

These two equations are derived by fitting a linear function to observed data at the reach scale as shown in Figure 2. Because we see it in our dataset, we then use it to simplify the hydraulic inverse problem by tying some of the parameters together, i.e. reducing the number of fitting parameters. Due to the spatial variation of α, γ, β and δ, we have to constrain 4*N of parameters (N is the number of cross sections). By using equations 11 and 12, we can halve the number of parameters.

We have expanded this section and appended the above derivations to the appendix A to clear up the doubts.

P7.L177:

> the residuals between model predictions and observations (i.e. water level and width);

It is not clear how the width is calculated in the model (as the model now only has flow-area without explicit width representations). How width and flow area can be related? Please explain in a detailed manner.

Reply: In the model, the width is calculated as the derivative of flow area with respect to depth. We have added this in section 3.3.

*"… $b_s$, $b_o$, $N_b$, $\sigma_b$ are simulated width (calculated as the derivative of flow area with respect to depth in the model), observed width, number of widths, and the uncertainty of observed width…"*

P7.L163

> Case study

I think it is better to explain the method (experimental setting) of the case study in the method section, and explain in a more detailed manner by moving descriptions from Supplement to main text. As HESS is a full paper journal, it is obviously better to write the minimum explanations on the experiment settings to follow the results.

Reply: Thank you. We have restructured the sections. The case study section includes study site, data and model setup, parameter calibration and calibration scenarios by integrating supplementary information.

P8.L182

> Figure 3

Please inform that the "km" indicates the distance from the upstream boundary. It is not clear which is

upstream/downstream from the current explanations.

Reply: Thank you for pointing this out. We use cross section number instead of chainage in the revised version. All numbers are indicated in the study map.

P8.L186:

"Figure 4" should be "Figure 3".

Reply: Revised.

P8.L191:

> the best match with the observed cross sections.

It is not so obvious that the calibration #3 is the best match, as we can observe some exceptions in some reaches. Please provide some objective measure (e.g. RMSE of fitting) to support this statement.

Reply: Thank you. We have added the RMSE and coverage to the figure. In terms of RMSE and coverage, calibration #3 is the best for most cross sections.

P9.L192:

> compared to less dense one (one per 5 km, Fig. S8) although high-resolution imagery (30 m) can provide plenty of width observations.

This explanation appears suddenly, and it is difficult to follow. I suggest that the authors describe how width observations is prepared in the method section (i.e. how it was sampled), and make an analysis on the sampling sensitivity in the discussion section (not in the case study section) to improve the readability.

Reply: We simply use the GEE app RivWidthCloud developed by Yang et al 2020. we have added a few sentences in the data section explaining the procedure for preparing width observations as below:

*"Widths are derived using the RivWidthCloud algorithm in Google Earth Engine (Yang et al., 2020). We used Landsat 5 and Landsat 8 images and selected images to avoid cloud cover and obtain an*

*even distribution in time. Specifically, if the river is cloud-free in a given image, it is selected regardless of the cloudiness of other parts. Images collected from December to early April are excluded. In total, 37 Landsat 5 images and 15 Landsat 8 images are used and provided 10022 individual width observations."*

P9.L196:

We can see the proposed method worked with a certain accuracy, but it is difficult to guess how much it is useful compared to other/previous methods. Please include some comparison with previous methods (e.g. comparison to the method by Jiang et al. 2019). Can you say the result is improved? Or can you say the result does not differ so much even though the new calibration requires less parameter numbers?

Reply: Yes, we have added the simulations using previous method (Jiang et al. 2019) to have a direct comparison with the results obtained using the method of this study. The results are comparable to what was achieved from previous one. See Figure R1 for a detailed comparison.

P10.L201:

> the accuracy of simulation is acceptable and comparable to what was achieved using a different approach (Jiang et al., 2019).

Please explain

1) What was the characteristics of the different method? (If not explained, this sentence has no meaningful

explanation). I suppose the Jiang et al 2019 calibration was "explicit cross-section and roughness parameters for each section" and thus have "more parameter numbers to be calibrated".

Reply: A clause has been added after this sentence to explain the method.

> *"The accuracy is comparable to what was achieved using a different approach, which simultaneously calibrates cross-sectional shape parameters and roughness (Jiang et al., 2019)."*

2) Please provide some numbers for subjective comparisons. For example, how the RMSE values are different in these two methods? Without the subjective analysis, I cannot get how the authors concluded that "accuracy is acceptable and comparable".

Reply: RMSEs of both methods are added on the figure. We have also added one more sentence to describe this.

> *"Overall, the accuracy of simulation is acceptable. RMSE is about 50 cm and 44 cm at Tonghe and Yilan stations, respectively. The accuracy is comparable to what was achieved using a different approach, which simultaneously calibrates cross-sectional shape parameters and roughness…"*

P11.L249:

> the relationship is generally independent of rivers

I think this is still the observation-based finding, and needs for some theoretical/geographical analysis should be discussed (in the discussion section).

Reply: Yes, this relationship is observed by fitting the data from 6 rivers. We have discussed this point in the discussion section as below:

> *"The power-law function relationships of flow area / conveyance and flow depth and the corresponding linearity approximation are confirmed in 6 rivers. One may argue that the*

*relationship may not be globally applicable due to the limited number of rivers to validate the relationships. We cannot rebut this argument without collecting a large sample of surveyed cross sections, which is difficult because of data access problems. However, the rivers we used are of diverse sizes (width ranging from a few meters to kilometers) and flow characteristics, and are from different climate zones (Arctic, Mediterranean, and Asian temperate climates). Therefore, we believe that the relationship holds globally, and we call for extensive validation using other rivers. Regarding the linear relationship between the flow area and conveyance curves for each cross section, it is understandable intuitively given that both flow area and conveyance are linearly related to the same variable, i.e. flow depth. At river reach scale, the strong linear relationships between α and γ, β and δ are empirical "*

Supplement:

Some contents are better to be moved to main text, such as Test S1, (a part of) Text S3 (except for sampling sensitivity experiments), Figure S5.

Reply: The supplementary information is fully integrated into the main text.

**Replies to the comments by referee #3**

Dear Guy J.-P. Schumann,

Thank you very much for reviewing our work. I have revised the title the objective according to your suggestions. Below we describe how we addressed your comments.

This paper describes the use of hydraulic geometry relationships to estimate river cross section geometry and the subsequent use of satellite altimetry and imagery to calibrate those parameters.

In fact this is a very well written paper and the methods are sound. The topic is also of interest and quite timely given the applicability of the method at the global scale.

However, I have one main point of concern. The methods the authors present, both the derivation of river geometry parameters and the calibration procedure with satellite data are not new at all, as the authors clearly allude to in their comprehensive and up-to-date literature review. There are now many studies looking at either one of those two approaches and indeed also combined.

In order for this paper to be publishable, I would therefore recommend to revise the title and not talk about a "new" scheme. In my opinion, the authors did a new river application study using existing methods.

Also, I strongly suggest the authors write a paragraph upfront in which they justify what is new in their work.

I gave this minor revisions because I think these two suggestions are relatively straightforward to implement in a revised version but, however, without these changes I would not recommend this current version to be published.

Reply: We thank the reviewer for acknowledging the contributions of our work presented in this study.

Following your suggestions, we have deleted the subtitle to avoid any overselling of this work. The current title is "Calibrating 1D hydrodynamic river models in the absence of cross-sectional geometry using satellite observations of water surface elevation and river width".

In the last paragraph, we explicitly state the novelty of this work as below:

> *"In order to reduce parameter correlation in hydraulic inverse problems, we put forward a method to parameterize and calibrate 1-D river models in a different way. Instead of roughness and geometry, flow area and conveyance curves as functions of flow depth are estimated in an inverse modeling workflow. In this way, only the dependence of area and conveyance on flow depth is estimated, regardless of the detailed channel shape and roughness. This paper illustrates this approach for the calibration of a 1D MIKE Hydro River model (DHI, 2017) to simulate WSE dynamics, using satellite observations of WSE and river width as observations. The novelty is to use power-law relationships between flow area / conveyance and flow depth in a hydraulic inversion without detailed cross-sectional data or assumption of any specific cross-sectional shape. Therefore, this approach is fundamentally different from previous studies, and provides an alternative way for hydrodynamic model calibration. "*

---

## Referee Report (RR1)

Dear Editor Albrecht Weerts,

Thank you for sending me the revised manuscript "Calibrating 1D hydrodynamic river models in the absence of cross-sectional geometry [...]" by Liguang Jiang et al. for review. The authors have responded to my comments in detail and improved their manuscript accordingly. They suggest easing hydrodynamic model calibration through a reparametrization of the SWE. The authors seem to have experience in model calibration and they demonstrate their method on hand of a suitable dataset. The method is mostly an incremental improvement on existing ones, though the approach contains several interesting details, such as the regularization approach, which will be insightful for the readers of HESS. It is in particular insightful to see that hydrodynamic models can be reliably calibrated with remotely sensed channel-width alone, i.e. without reference depth, which is of importance for modelling ungauged basins.

While I commend the manuscript, its language should be thoroughly revised before publication. I provide an extensive list of suggestions below. It would be considerate when the Danish co-authors supported the first author with that respect before submitting future manuscripts.

Kind regards,

Reviewer

**Minor**

Structure: I recommend moving subsection 3.3 "Parameter calibration" forward at the end of section 2, as it is part of the methodology.

Appendix A: It is nice to see the parameter relations worked out. It would be insightful to show how the physical parameters are related, and thus could be recovered, from the calibration parameters. For example $\alpha = \log(w/d_0)$, where $d_0$ is a reference depth and $1 < \beta < 2$, which are the limits for a rectangular and triangular cross-sections. Similar simple relations exist for $K$. This is will be helpful for choosing suitable start values for the optimization and verifying the result.

Code and data availability: It would be considerate if the authors made their code publicly available so that others can easily apply the method in their studies.

**Suggested clarifications**

"two variables (Q and d) and three parameters $(A, S_0,$ and $S_f$ )"
$\rightarrow$ Two variables (Q and d) and three unknown values $(A, S_f$ and $S_0)$, which are functions of further parameters as specified below.

"K(d) is much more sensitive than A(d)."
$\rightarrow$ The calibration is much more sensitive to K than to A.

"parameters [...] in addition to bed slope $S_0$ (calculated from $Z_0$)"
$\rightarrow$ "parameters [...] in addition to the bed level $Z_0$, from which the bed slope $S_0$ is calculated."

Some more information would be insightful here. Are the cross-sections of the hydrological model, or of the validation data? If they are of the model, are the parameters linearly interpolated between the sections? In which interval are the cross-sections placed?

Insert the missing sum signs in front of the brackets which are squared.

Is the value later reported as RMSE the "misfit" or only the standard deviation between the 10 calibration runs?

238-243 We use the LM: algorithm [...] to optimize the objective function.
$\rightarrow$ We iteratively optimize the objective function (equation. 17) with the Levenberg-Marquardt (LM) algorithm (Marquardt, 1963) combined with Broyden's rank-one update to approximate the Jacobian (Broyden, 1965, Madsen et al. 2004). We use an implementation of the method provided by the Immoptibox toolbox (Nielsen and Völcker, 2010).

**Suggested textual improvements**

- All equations: End with dot or comma, as the equations are part of the sentence.

Scarcity/inaccessibility $\rightarrow$ Scarcity and inaccessibility geometry has commonly been approximated using $\rightarrow$ geometry is commonly approximated by

Simultaneous [...] $\rightarrow$ Some explanation is missing before this sentence. For example: Hydrological model calibration requires both the determination of parameters for roughness and cross-section geometry.

,20 power-law functions $\rightarrow$ power-laws remove "and they are found to be linearly [...]", this is already implied by the power-law and thus an unnecessary tautology has been → is different → multiple require detailed → require a detailed by cross-sectional → by a cross-sectional surveyed [. . . ] geometry → the surveyed geometry problem facing the scientific community → problem which the scientific community faces used a uniform shape . . . → used a cross-section geometry which did not vary along the channel.

Here → Here, to simulate → for simulating morphology → roughness

When calibrating [. . . ] → Parameters of channel geometry and roughness are highly correlated during calibration.

will be effective, not only representing → effectively represent compensating → compensate remove "as observations"

(2) $S_f \rightarrow S_f(d)$

is chainage, i.e. the distance → is the distance

To effect solution → to solve for bathymetry → the bathymetry

Friction slope → The friction slope provide → are power function relationships → power-laws

$Z \rightarrow Z_0$

different → several having a wide range of river width (three orders of magnitude). The width ranges between the rivers over three orders of magnitude.

remove "readily"

power-law function → power-law

,123,124 Manning's number → Manning's coefficient and representative of large rivers worldwide. → one of the largest rivers in the world.

two rivers merge into one, called Songhua → two tributaries merge to form the Songhua River emptying → draining remove "main"

The reason we selected this reach is twofold → The reasons why we selected this reach are twofold:

drains → flows at downstream → at the downstream end set → set up

Daily → The daily remove "new"

entirely different → unique derived → extracted to avoid → avoiding uncertainty → root mean square error?

log depth → log Depth small depth → small depths very wide range of RMSE → very large RMSE?

WSE → the WSE

RMSE → The RMSE

which is not new [. . .] → which goes back to Chow 1959.

and the relationship is generally independent of rivers → and applicable for a wide range of rivers.

no explicit consideration of roughness and channel geometry are needed to solve for WSE → the channel geometry and roughness do not have to be explicitly known to determine the WSE.

WSE → the WSE

remove the qualifier "fairly"

By referring [. . . ] → Our method performs comparably to existing ones which use conventional parametrization and calibration approaches.

this approach → our approach we can get → we get

Taking log transformation → Taking the logarithm that leaves → we have

---

## Author Response (AR2)

**Replies to the comments by referee #1**

Dear Editor Albrecht Weerts,

Thank you for sending me the revised manuscript "Calibrating 1D hydrodynamic river models in the absence of cross-sectional geometry [. . . ]" by Liguang Jiang et al. for review. The authors have responded to my comments in detail and improved their manuscript accordingly. They suggest easing hydrodynamic model calibration through a reparametrization of the SWE. The authors seem to have experience in model calibration and they demonstrate their method on hand of a suitable dataset. The method is mostly an incremental improvement on existing ones, though the approach contains several interesting details, such as the regularization approach, which will be insightful for the readers of HESS. It is in particular insightful to see that hydrodynamic models can be reliably calibrated with remotely sensed channel-width alone, i.e. without reference depth, which is of importance for modelling ungauged basins.

While I commend the manuscript, its language should be thoroughly revised before publication. I provide an extensive list of suggestions below. It would be considerate when the Danish co-authors supported the first author with that respect before submitting future manuscripts.

Kind regards,

Reviewer

Reply: Thank you very much for taking the efforts to have a careful read and comment on our revised manuscript. Below, we report how we revised the text following your suggestions.

**Minor**

Structure: I recommend moving subsection 3.3 "Parameter calibration" forward at the end of section 2, as it is part of the methodology.

Reply: Fully agree. We have revised it accordingly.

Appendix A: It is nice to see the parameter relations worked out. It would be insightful to show how the physical parameters are related, and thus could be recovered, from the calibration parameters. For example, $\alpha = \log(w/d_0)$, where $d_0$ is a reference depth and $1 < \beta < 2$, which are the limits for a rectangular and triangular cross-sections. Similar simple relations exist for K. This is will be helpful for choosing suitable start values for the optimization and verifying the result.

Reply: At reach scale, the parameters ($\alpha$, $\beta$, $\gamma$, and $\gamma$) are empirical. Even at-a-station, $\alpha$ and $\beta$ range from 0 to 2.5 and 1 to 3.5, respectively, for the six rivers we investigated. The values for rectangular and triangular cross-sections do not necessarily stand for the limits. Figure 2 as shown below clearly reveals that the value of $\beta$ can be larger than 2.

[Figure]

Figure 2. Linear relationship between $\alpha/\beta$ and $\gamma/\delta$. Each dot represents one cross-section of a certain river. Dots of the same color are from the same river. Manning's coefficient for each cross-section is randomly generated between 0.015 and 0.05.

Code and data availability: It would be considerate if the authors made their code publicly available so that others can easily apply the method in their studies.

Reply: Sure, we have made it available on Zenodo (https://zenodo.org/record/5555472). We have noted this in the code and data availability section.

**Suggested clarifications**

"two variables (Q and d) and three parameters (A; S0, and Sf )" → Two variables (Q and d) and three unknown values (A; Sf and S0), which are functions of further parameters as specified below.

Reply: Revised.

"K(d) is much more sensitive than A(d)." → The calibration is much more sensitive to K than to A.

Reply: Revised.

"parameters [. . . ] in addition to bed slope S0 (calculated from Z0)" → "parameters [. . . ] in addition to the bed level Z0, from which the bed slope S0 is calculated."

Reply: Revised.

Some more information would be insightful here. Are the cross-sections of the hydrological model, or of the validation data? If they are of the model, are the parameters linearly interpolated between the sections? In which interval are the cross-sections placed?

Reply: These cross-sections are used for setting up the hydrodynamic model instead of real-world data. There are in total 23 cross-sections over a span of 433 km as illustrated in Figure 3. Yes, cross-sections are linearly interpolated along chainage in MIKE Hydro River.

To make it clearer, we have revised the text as *"…The first step is to define the river network, cross sections, and boundary conditions. The river network is set up using the center line of the reach,*

*while 23 cross sections are equally distributed along the 433 km reach as in Jiang et al. (2019). Daily discharge at Harbin hydrometric station is used as the upstream boundary while a uniform flow depth rating curve is set as downstream boundary …"*

Insert the missing sum signs in front of the brackets which are squared.

Reply: Revised.

Is the value later reported as RMSE the "misfit" or only the standard deviation between the 10 calibration runs?

Reply: The RMSEs are simply calculated based on model-simulated WSE and in-situ or satellite altimetry derived WSE. And yes, in Figure 5, we calculated the statistics using the 10 calibration runs. This has been added in the caption of Figure 5.

238-243 We use the LM: algorithm [. . . ] to optimize the objective function. → We iteratively optimize the objective function (equation. 17) with the Levenberg-Marquardt (LM) algorithm (Marquardt, 1963) combined with Broyden's rank-one update to approximate the Jacobian (Broyden, 1965, Madsen et al. 2004). We use an implementation of the method provided by the Immoptibox toolbox (Nielsen and Völcker, 2010).

Reply: Revised.

**Suggested textual improvements**

All equations: End with dot or comma, as the equations are part of the sentence.

Reply: Revised.

Scarcity/inaccessibility → Scarcity and inaccessibility

Reply: Revised.

geometry has commonly been approximated using → geometry is commonly approximated by

Reply: Revised.

Simultaneous [. . . ] → Some explanation is missing before this sentence. For example: Hydrological model calibration requires both the determination of parameters for roughness and cross-section geometry.

Reply: Revised.

18, 20 power-law functions → power-laws

Reply: Revised.

remove "and they are found to be linearly [. . . ]", this is already implied by the power-law and thus an unnecessary tautology

Reply: The power-laws are directly applied at each cross-section. The linearity that exists at reach scale is not explicit from the cross-sectional power-laws. We would like to keep this point in the abstract.

has been → is

Reply: Revised.

different → multiple

Reply: Revised.

require detailed → require a detailed

Reply: Revised.

by cross-sectional → by a cross-sectional

Reply: Revised.

surveyed [. . . ] geometry → the surveyed geometry

Reply: Revised.

problem facing the scientific community → problem which the scientific community faces

Reply: Revised.

used a uniform shape . . .--> used a cross-section geometry which did not vary along the channel.

Reply: Revised.

Here → Here,

Reply: Revised.

to simulate → for simulating

Reply: Revised.

morphology → roughness

Reply: Revised.

When calibrating [. . . ] → Parameters of channel geometry and roughness are highly correlated during calibration.

Reply: Here we emphasize the simultaneous calibration of geometry parameters and roughness parameters. We think it is better to use the original expression.

will be effective, not only representing → effectively represent

Reply: The suggested expression is not informative. We keep our original expression.

compensating → compensate

Reply: "not only representing [...] but also compensating" We do not think it is inappropriate.

remove "as observations"

Reply: Revised.

(2) Sf → Sf (d)

Reply: Revised.

is chainage, i.e. the distance → is the distance

Reply: We keep the "chainage" term as it is used later in the text.

To effect solution → to solve for

Reply: Revised.

bathymetry → the bathymetry

Reply: Revised.

Friction slope → The friction slope

Reply: Revised.

provide → are

Reply: Revised.

power function relationships → power-laws

Reply: Revised.

Z → Z0

Reply: Revised.

different → several

Reply: Revised.

having a wide range of river width (three orders of magnitude). →The width ranges between the rivers over three orders of magnitude.

Reply: Revised.

remove "readily"

Reply: Revised.

power-law function → power-law

Reply: Revised.

122,123,124 Manning's number → Manning's coefficient

Reply: Revised.

and representative of large rivers worldwide. → one of the largest rivers in the world.

Reply: Revised.

two rivers merge into one, called Songhua → two tributaries merge to form the Songhua River

Reply: Revised.

emptying → draining

Reply: Revised.

remove "main"

Reply: Revised.

The reason we selected this reach is twofold → The reasons why we selected this reach are twofold:

Reply: Revised.

drains → flows

Reply: Revised.

at downstream → at the downstream end

Reply: Revised.

set → set up

Reply: Revised.

Daily → The daily

Reply: Revised.

remove "new"

Reply: Revised.

entirely different → unique

Reply: We intend to emphasize the "difference" instead of its uniqueness.

derived → extracted

Reply: Revised.

to avoid → avoiding

Reply: Revised.

uncertainty → root mean square error?

Reply: Not necessary. But we can use rmse to represent the uncertainty or error of data.

log depth → log Depth

Reply: Revised.

small depth → small depths

Reply: Revised.

very wide range of RMSE → very large RMSE?

Reply: Here, we mean that the values are very spread instead of very close to each other. We revised as "*the RMSE values of calibrations #2a and #2b are very spread (i.e. a wide range)*".

WSE → the WSE

Reply: Revised.

RMSE → The RMSE

Reply: Revised.

which is not new [. . . ] → which goes back to Chow 1959.

Reply: Revised.

and the relationship is generally independent of rivers → and applicable for a wide range of rivers.

Reply: Revised.

no explicit consideration of roughness and channel geometry are needed to solve for WSE → the channel geometry and roughness do not have to be explicitly known to determine the WSE.

Reply: Revised.

WSE → the WSE

Reply: Revised.

remove the qualifier "fairly"

Reply: Revised.

By referring [. . . ] → Our method performs comparably to existing ones which use conventional parametrization and calibration approaches.

Reply: Revised.

this approach → our approach

Reply: Revised.

we can get → we get

Reply: Revised.

Taking log transformation → Taking the logarithm

Reply: Revised.

that leaves → we have

Reply: Revised.

**Replies to the comments by referee #2**

I appreciate the authors to largely improve the manuscript following my comments/concerns on the previous version. The significance of the proposed work is now explained appropriately, and the comparison to previous methods (Jiang et al 2019) enhance the reliability of the proposed method.

I only have a minor technical comments. In Figure 6, what "Simulation S1" mean is not clear (what S1 stands for? it is explained only in the maintext). I suggest to modify the caption as "(i.e. simultaneous calibration of roughness and cross-sectional shape parameters; S1 simulation in Jiang et al 2019), in order to explain what S1 means.

Reply: Thank you for the positive comments. We have revised the caption according to your suggestion.